# Cryo-EM reveals the complex architecture of dynactin's shoulder region and pointed end

Clinton K Lau[1] (ID), Francis J O'Reilly[2] (ID), Balaji Santhanam[1] (ID), Samuel E Lacey[1] (ID), Juri Rappsilber[2] (ID) & Andrew P Carter[1,*] (ID)

## Abstract

**Dynactin is a 1.1 MDa complex that activates the molecular motor dynein for ultra-processive transport along microtubules. In order to do this, it forms a tripartite complex with dynein and a coiled-coil adaptor. Dynactin consists of an actin-related filament whose length is defined by its flexible shoulder domain. Despite previous cryo-EM structures, the molecular architecture of the shoulder and pointed end of the filament is still poorly understood due to the lack of high-resolution information in these regions. Here we combine multiple cryo-EM datasets and define precise masking strategies for particle signal subtraction and 3D classification. This overcomes domain flexibility and results in high-resolution maps into which we can build the shoulder and pointed end. The unique architecture of the shoulder securely houses the p150 subunit and positions the four identical p50 subunits in different conformations to bind dynactin's filament. The pointed end map allows us to build the first structure of p62 and reveals the molecular basis for cargo adaptor binding to different sites at the pointed end.**

**Keywords** cargo adaptors; conservation; cryo-electron microscopy; dynactin; protein complex

**Subject Categories** Cell Adhesion, Polarity & Cytoskeleton; Structural Biology

**The EMBO Journal (2021) 40: e106164**

## Introduction

Dynactin is a large, multi-subunit co-activator of the molecular motor cytoplasmic dynein 1. It is required for long-range transport along microtubules in many animals and fungi (Reck-Peterson *et al*, 2018) (Fig 1A). Dynactin is built around an Arp1/actin filament, which is capped by pointed and barbed-end complexes (Schafer *et al*, 1994; Eckley *et al*, 1999; Imai *et al*, 2014; Chowdhury *et al*, 2015; Urnavicius *et al*, 2015) (Fig 1B). On the side of the filament sits a shoulder domain from which the ~ 75 nm-long p150^Glued (*DCTN1*, hereafter referred to as p150) projection extends

(Schafer *et al*, 1994; Urnavicius *et al*, 2015). Dynein binds dynactin in the presence of coiled-coil cargo adaptors, such as BICD2, BICDR1, and Hook3 to form a highly processive motor complex (McKenney *et al*, 2014; Schlager *et al*, 2014a). Dynein contacts the Arp1 filament via its heavy chain (Urnavicius *et al*, 2015; Urnavicius *et al*, 2018), and the p150 N terminus via its intermediate chain (Karki & Holzbaur, 1995; Vaughan & Vallee, 1995). Coiled-coil adaptors make interactions along dynactin's filament and pointed end and bind dynein's heavy chain and light intermediate chain (Schroeder *et al*, 2014; Urnavicius *et al*, 2015; Gama *et al*, 2017; Lee *et al*, 2018; Urnavicius *et al*, 2018).

Despite multiple structures containing dynactin (Urnavicius *et al*, 2015; Urnavicius *et al*, 2018), neither the shoulder nor the pointed end has yet been resolved at high resolution. The shoulder consists of the C-termini of two p150 subunits, four copies of p50 (*DCTN2*) and two p24s (*DCTN3*) (Eckley *et al*, 1999). The N-termini of the p50s bind the Arp1 filament and act as molecular rulers to determine its length (Melkonian *et al*, 2007; Cheong *et al*, 2014; Urnavicius *et al*, 2015). Previous studies showed that the shoulder contains long three-helical bundles with a twofold pseudo-symmetry (Urnavicius *et al*, 2015). However, due to the limited resolution, it was not possible to assign individual subunits. Key outstanding questions include how the C-termini of p150 are embedded into the shoulder, how the four p50 subunits organize into a structure with twofold symmetry, and how their N-termini project to correctly bind the filament.

The pointed end is important for binding dynein-dynactin cargo adaptors (Zhang *et al*, 2011; Yeh *et al*, 2012; Urnavicius *et al*, 2015; Gama *et al*, 2017; Qiu *et al*, 2018; Urnavicius *et al*, 2018). It consists of four subunits: actin-related protein 11 (Arp11, *ACTR10*); p62 (*DCTN4*); p25 and p27 (*DCTN5* and *DCTN6*). Previous maps were sufficient to build Arp11 and place, but not assign, models of p25 and p27 (Yeh *et al*, 2013; Urnavicius *et al*, 2015; Urnavicius *et al*, 2018). It was not possible to model p62 due to poor density and lack of structural homologs. Using the previous dynein tail-dynactin-BICD2 structures (Urnavicius *et al*, 2015), structural modeling with molecular dynamics predicted p25 residues to bind to all coiled-coil adaptors (Zheng, 2017). However, subsequent structures revealed that Hook3 and BICDR1 in fact contact different regions of the pointed end (Urnavicius *et al*, 2018). The lack of high

1 Structural Studies Division, MRC Laboratory of Molecular Biology, Cambridge, UK
2 Bioanalytics, Institute of Biotechnology, Technische Universität Berlin, Berlin, Germany
   *Corresponding author. E-mail: cartera@mrc-lmb.cam.ac.uk

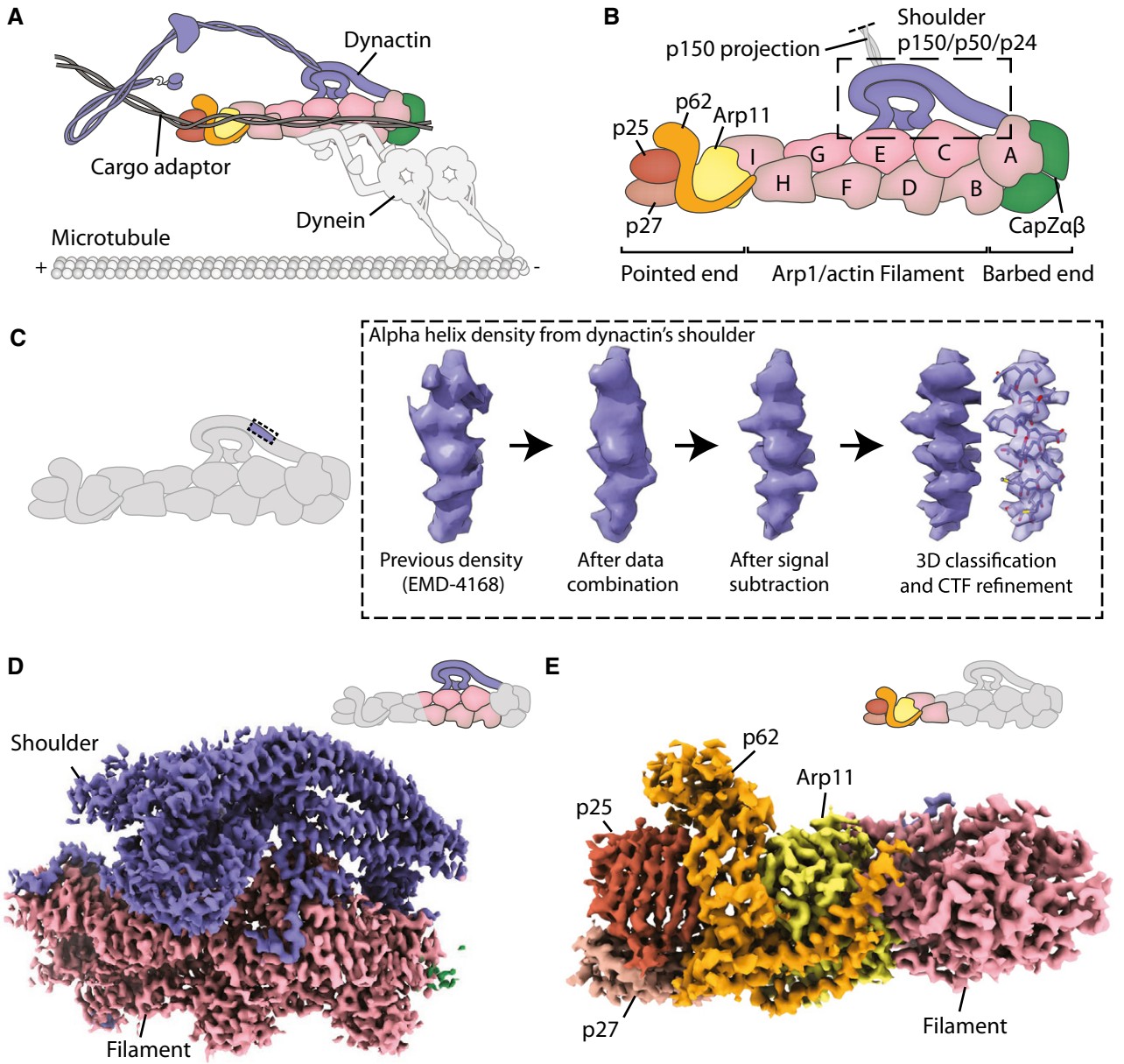

**Figure 1.  High-resolution maps of dynactin's shoulder and pointed end.**

A  Schematic showing a dynein–dynactin–adaptor complex on microtubules.

B  Schematic showing the domain architecture of dynactin.

C  Density improvements during processing. For each step, density shown is taken from the same p50 arm helix in the shoulder.

D  Map of the shoulder region. Density of the shoulder is colored in blue.

E  Map of the pointed end including Arp 11 (yellow); p62 (orange); p25 (brown); and p27 (light brown).

resolution in these regions means that it is currently unclear which pointed end residues interact with the different cargo adaptors.

To overcome the flexibility within dynactin, we combined multiple cryo-EM datasets of different dynactin-containing complexes and developed a precise masking strategy for signal subtraction. In combination with other recent advances in cryo-EM data processing, this allowed us to produce a 3.8 Å map of dynactin's shoulder and a 4.1 Å map of the pointed end. We find that the p150 C-termini are securely anchored into the shoulder by making extensive interactions with other subunits. The p50 subunits are asymmetrically arranged in four unique conformations to position their N-termini correctly to bind to dynactin's filament. At the pointed end of dynactin, we build an atomic model of p62 and identify the residues involved with cargo adaptor binding. We also resolve the pointed end residues that interact with the p150 projection when it folds back to contact dynactin. We find that in this conformation p150 overlaps with all adaptor-binding sites, suggesting that it acts to inhibit dynactin's interactions with cargo adaptors.

# Results

## Determination of high-resolution structures of dynactin's shoulder and pointed end

One cause of limited resolution in previous dynactin structures was flexibility that smeared the density in peripheral regions. Particle signal subtraction can overcome this by computationally subtracting density around regions of a protein complex that move as a rigid body, permitting further refinement to higher resolution (Bai *et al*, 2015). In our previous structure of dynein tail-dynactin-BICDR1 (TDR), this approach allowed us to build an atomic model of the dynein tails (Urnavicius *et al*, 2018). Using that dataset, we first attempted to implement the same strategy for dynactin's shoulder and pointed end. However, signal subtraction on these regions using the TDR dataset alone did not produce maps of sufficient quality to build an atomic model. To overcome this, we decided to increase our particle number, then use signal subtraction with mask optimization, 3D classification, CTF refinement, and particle recentering to increase the resolution.

We increased our dataset size by combining data from our previous dynactin (Urnavicius *et al*, 2015), TDR and dynein tail-dynactin-Hook3 (TDH) structures (Urnavicius *et al*, 2018), and by incorporating new TDH data (Fig EV1 and Appendix Table S1). For the TDR and TDH datasets, we used signal subtraction to remove density for the dyneins and cargo adaptors. Using the resulting dynactin particles, we focused on either the shoulder or pointed end and performed signal subtraction and local refinement for each region (Figs 1C and EV1). To determine the best possible mask for this process, we first tested a broad range of masks to identify regions that could be refined to higher resolution. We proceeded with masks that gave the maps containing the best density.

For the shoulder, we next optimized the mask. This was accomplished by testing the mask using focused refinement without signal subtraction. We examined the boundaries of the output map to identify further density to include or exclude (Appendix Fig S1). Specifically, we adjusted the shoulder mask to exclude Arp1 subunit A, which was slightly flexible relative to the rest of the map, and include parts of Arp1 subunits F and G, which were well-resolved, but outside our original mask. Then we used the optimized mask for signal subtraction. This resulted in better density throughout the map compared to non-optimized versions, allowing us to build a more complete structure. CTF refinement followed by 3D classification for the shoulder region further improved the density (Figs 1D and EV1, and Appendix Fig S2).

For the pointed end, we used a relatively large mask for initial signal subtraction. We simultaneously recentered our particles, using the feature introduced in RELION 3.1 (Zivanov *et al*, 2020), as this region is located far from the center of the particle. Recentering permits more meaningful priors for refinement, meaning it reduces the errors in assignment of rotations and offsets of the particles. This allowed the visualization of β strands in p62 for the first time. After the first round of subtraction, the mask was further optimized before a second round of signal subtraction using the same strategy as described above. This was followed by 3D classification and further refinement (Figs 1E and EV1, and Appendix Fig S3).

Using the new maps, we could build models for the shoulder (Figs 2–4) and pointed end (Fig 5). To validate our structures, we crosslinked dynactin with bis(sulfosuccinimidyl)suberate (BS3) and identified the crosslinked residues using cross-linking mass spectrometry (PRIDE dataset PXD020084). 527 crosslinks were identified with a false discovery rate of 2% and were compared against our structure (Appendix Fig S4A-C). 246 of the crosslinked residues pairs are in structurally ordered regions in dynactin. A further 59 pairs have one or both residues contained in short disordered loops that can be modeled. These 305 crosslinks satisfy the maximum theoretical length of BS3, 30 Å ($C_\alpha$-$C_\alpha$) (Appendix Fig S4D). Of the 18 overlength crosslinks, 7 can be explained by minor structural flexibility. The other 11 are incompatible with our structure, consistent with our false discovery rate. The remaining crosslinks involve at least one residue in long disordered loops (35 crosslinks) or within the p150 projection (169 crosslinks).

## The complex structure of dynactin's shoulder

Dynactin's shoulder can be split into two subdomains that stack on top of each other, with the lower subdomain making more interactions with dynactin's filament (Fig 2A). As previously described (Urnavicius *et al*, 2015), each subdomain has a long three-helical arm and two shorter helical domains, referred to as the hook and paddle (Fig 2B and C). These two subdomains were seen to be linked by a dimerization domain. However, without higher resolution data it was impossible to assign which part of the structure belonged to which of the subunits: p50, p24, and the C-terminal domain of p150 (Appendix Figs S5 and S6).

Our new maps are of sufficient quality to allow us to now build a full model of the shoulder. We have sidechain-resolution density covering the dimerization domain, hook, paddle, and the majority of the arm for at least one of the subdomains (Appendix Fig S7). The only region lacking sidechain density in both subdomains is the middle of the arms (Appendix Fig S7, asterisk). The maps have good connectivity at lower threshold allowing us to be confident we can trace the complete paths of each subunit. The final model is supported by our cross-linking mass spectrometry data, with 167 crosslinks within the shoulder (Appendix Fig S4B). Our structure shows that each subdomain contains two p50s, one p24, and one p150 and that their arrangement is equivalent in both subdomains. Here we describe the lower subdomain, shown in Fig 3A colored by subunit, with the p50 subunits in red (p50-A) and pink (p50-B), the p24 subunit in yellow, and the p150 in purple.

The p150 subunit enters the shoulder around residue 1096 (Fig EV2A). Consistent with this, the region of p150 close to this point makes multiple crosslinks with the parts of p50 and p24 in the middle of the arm (Appendix Fig S8A). Residues 1096–1140 run along the arm, making contacts with both p50 and p24 subunits. The next section of p150 forms a helical hairpin that accounts for two-thirds of the hook (Fig 3A). The polypeptide chain then unexpectedly folds into the dimerization domain, contributing two β-strands and one α-helix (residues 1253–1286, Fig 3A and B). The way in which p150 is intricately interwoven with other shoulder subunits strongly suggests that it is an obligate part of dynactin's shoulder (Fig 3A).

The region of p50 that is embedded in the shoulder includes residues 100–405 (Fig 3D, and Appendix Fig S5). This part of the protein contains 8 α-helices (H1–H8) and one β-strand (S1) and is present in two copies per subdomain. The C-terminal portions of the

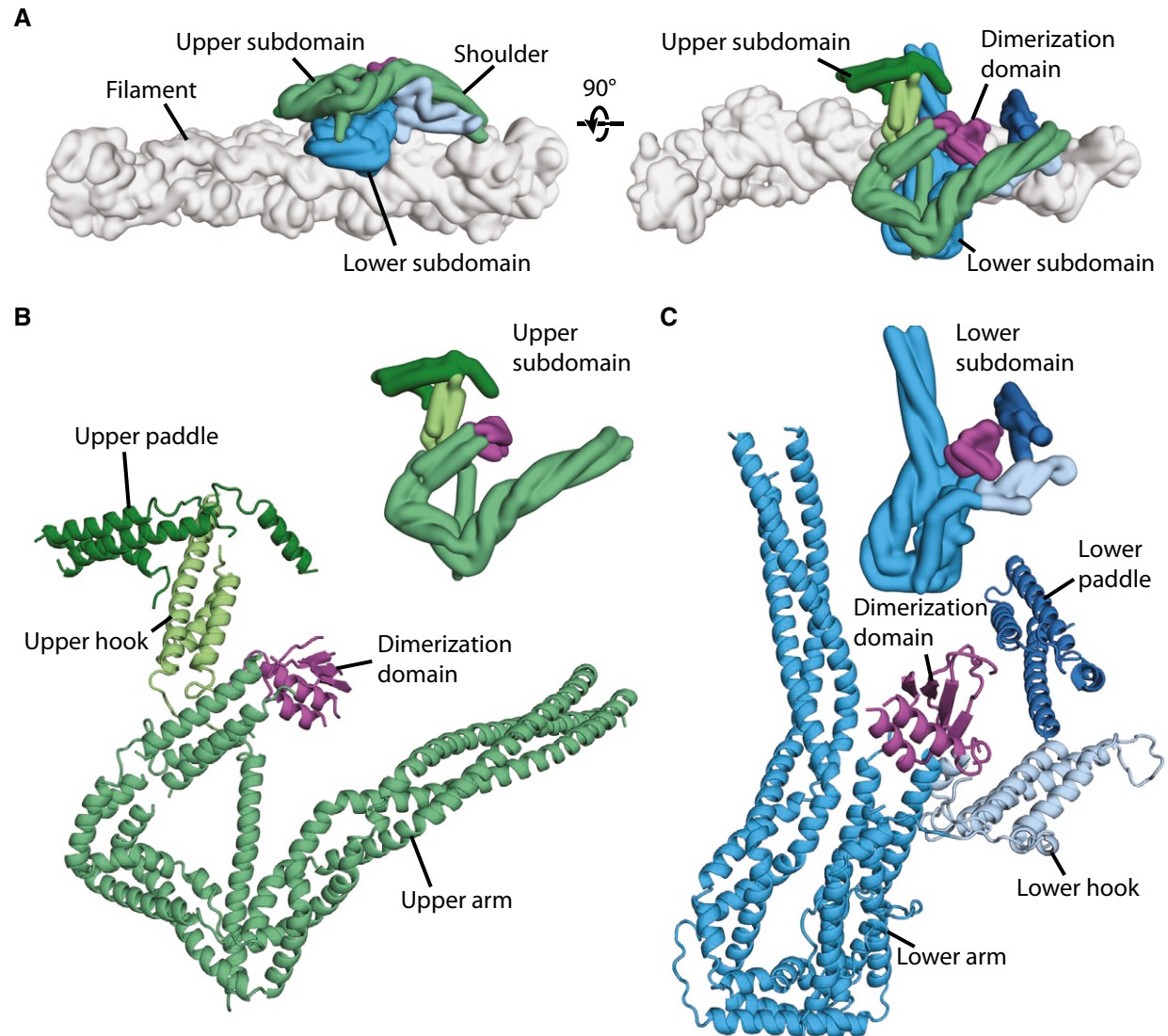

**Figure 2. Structure of dynactin's shoulder.**

A   Gaussian surface rendering of dynactin showing the arrangement of the upper subdomain (greens) and the lower subdomain (blues) of the shoulder on the filament (gray). Different features are shown in different color shades, and the dimerization domain is colored magenta.
B   Upper subdomain of the shoulder, shown in ribbon (main) or surface representation (inset), with the arm, hook, and paddle colored in shades of greens.
C   Lower subdomain of the shoulder, shown in ribbon (main) or surface representation (inset), with the arm, hook, and paddle colored in shades of blue.

two p50s, containing helices H4–H8, are equivalent in structure and are part of the long arm (Fig 3A). In contrast, the more N-terminal parts (H1–3 and S1) diverge and contribute differently to the dimerization domain, the hook and paddle regions (Fig 3A).

The C-terminal portions of p50 form a three-helix bundle with the entirety of p24 to make up the long arm. All three chains run C- to N-terminal from the distal tip of the arm back toward the dimerization domain (Figs 3A and EV2B), with equivalent residues in the two p50 subunits approximately alongside each other. In the middle section of the arm, we see crosslinks between p50 and p24 (Appendix Fig S8A, dashed lines), consistent with our structure. Surprisingly, in the section of the arm proximal to the dimerization domain, the helical bundle breaks, twists by 120°,

then reforms (Fig 3C). In p24, this break is spanned by a short loop (residues 23–32) that is visible in our structure (Fig 3C and D, and Appendix Fig S8B). In p50, there are much longer loops (residues 243–260, Fig 3C and D). In one p50 copy, p50-A, this long loop is ordered (Fig 3C, solid red line, and Appendix Fig S8B), packing against the p150 subunit and hence contributing to the stability of the structure.

Whereas p24 is wholly contained in the arm, the two p50s connect the arm to the dimerization domain. In p50-A, the single β-strand (S1) and its preceding helix (H3) both contribute (Fig 4A and B), with the β-strand positioned at the dimer interface (Fig 3B). In p50-B, only the β-strand is involved (Fig 4A and C), sitting between the two β-strands derived from p150 (Fig 3B).

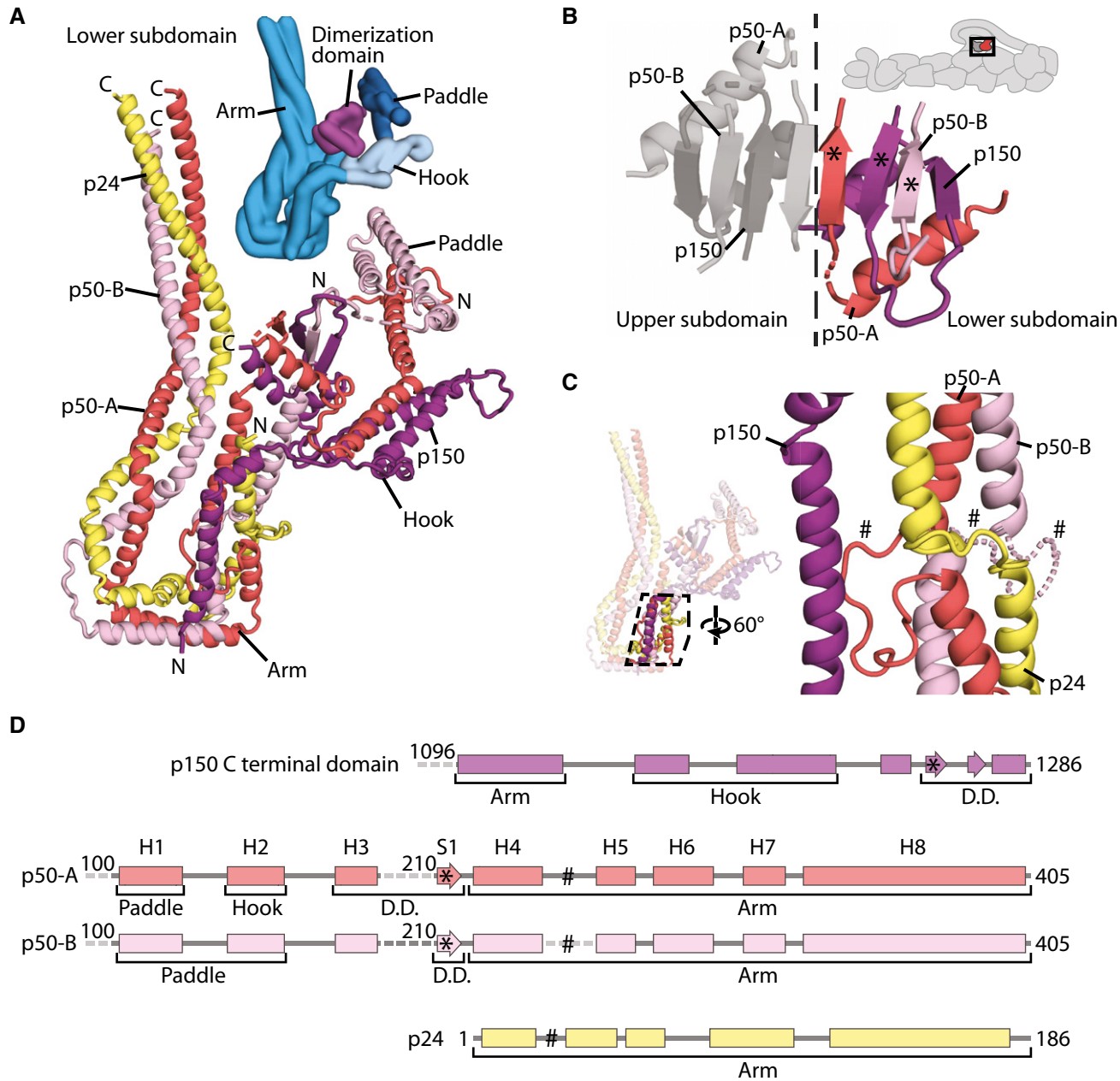

**Figure 3. Arrangement of dynactin's shoulder subunits.**

A  The structure of the lower subdomain from the shoulder showing the organization of p150 (purple), p24 (yellow), and the two copies of p50 (p50-A in red, p50-B in pink). The N- and C-termini of each chain are shown. A Gaussian surface rendering of the lower subdomain is colored by feature (blues and magenta, inset).

B  The dimerization domain of the shoulder. Subunits from the lower subdomain are colored, with p50-A in red, p50-B in pink, and p150 in purple. * marks equivalent features in (D).

C  Helical bundle break in the lower subdomain arm, showing how the three helices in the arm break and reform after a twist. p24 (yellow) forms a short loop at the break, whereas p50 forms a longer loop, which is ordered in p50-A (red), and disordered in p50-B (pink), modeled by a dotted line. # marks equivalent features in (D).

D  Secondary structure diagrams for the segments of p150, p50, and p24 within dynactin's shoulder. Helices H1-8 and S1 are labeled for p50-A and p50-B.

In the hook and the paddle, the more N-terminal portions of the two p50s (helices H1–H3) fold into different conformations (Fig 4D and E). p50-A contributes its H2 helix to the hook region and its H1 helix to the paddle. In p50-B, the H3 helix lies near the hook region. Here, it contacts the arm of the other subdomain, playing a key role in holding the two subdomains together (Appendix Fig S8C). Since H3 of p50-A is in the dimerization domain, this means that the two H3 helices in each subdomain are 60 Å apart. This radically different arrangement is facilitated by a long loop between the S1 β-strand and the H3 helix. This loop is flexible in p50-A, but can be

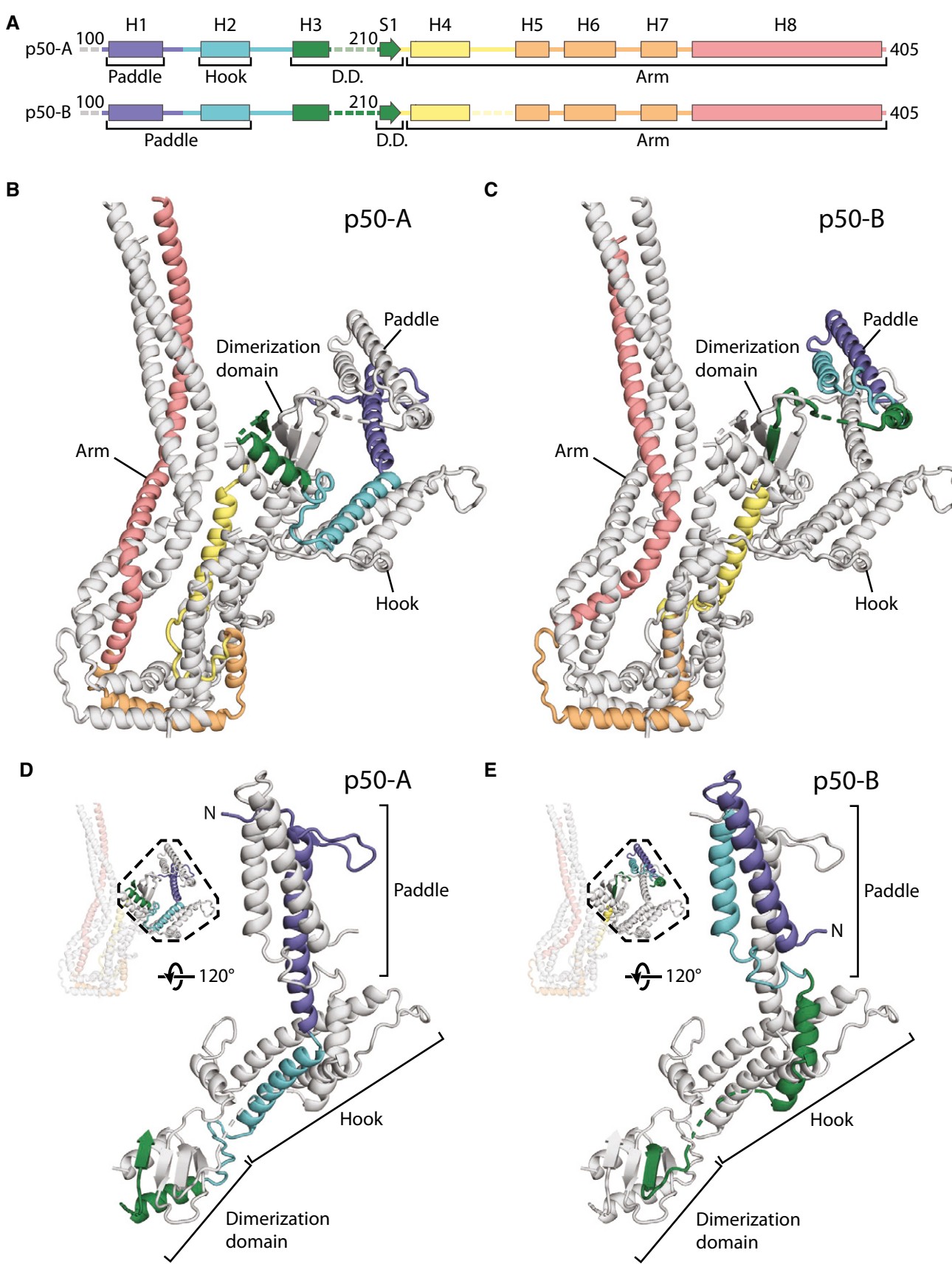

**Figure 4.**

**Figure 4. Alternative conformations of p50-A and p50-B.**

A       Secondary structure diagram of p50-A and p50-B colored in rainbow from N- to C-termini in the shoulder. Helices H1-8 and S1 are labeled for p50-A and p50-B.
B, C    Ribbon diagram of the lower subdomain of the shoulder showing the path of p50-A (B) and p50-B (C), colored as in (A).
D, E    The N-terminal halves of p50-A (D) and p50-B (E) adopt different conformations in each subdomain (lower subdomain shown), colored as in (A).

seen in low threshold maps for p50-B where it is pulled almost taut (Fig 4A and E, and Appendix Fig S8C). The H1 and H2 helices of p50-B form a helical hairpin that contributes to the paddle region. The different arrangement of the two p50 subunits in our model is validated by our mass spectrometry data. The p50 H2 and H3 helices both crosslink to the same region of p24, near its C-terminus (Appendix Fig S8D). Our structure explains these crosslinks (Appendix Fig S8E), with p50-A H2, and p50-B H3 near these p24 residues. In contrast, H2 from p50-B and H3 from p50-A are both over 50 Å from this site, which is too far for crosslinks to form (Appendix Fig S8E).

As a consequence of the different p50 conformations, the extended N-termini of the two p50s (residues 1–100) within each subdomain project from opposite sides of the paddle domains (Fig 4D and E). For the lower subdomain, we can connect p50-A to the N terminus that contacts Arp1 subunits B and D in the filament (ER-3 from Urnavicius *et al*, 2015). At lower threshold, we can connect p50-B to the N terminus on the other side of the filament that interacts with Arp1 subunits A, C, and E (ER-1 from Urnavicius *et al*, 2015). The connections from the upper subdomain to the filament are weak but based on proximity we predict that p50-A connects to Arp subunits G and I (ER-2), and p50-B to subunit F (ER-4). Our structure shows that the asymmetry in the shoulder results in each of the four N-termini to be presented uniquely, in order to interact with all eight Arp1 subunits (Urnavicius *et al*, 2015).

**The assembly of dynactin's pointed end complex**

In previous structures of dynactin's pointed end, only Arp11 showed density for side chains (Urnavicius *et al*, 2015). In our new map, we can now build structures of p62, p25, and p27 and determine how they interact (Fig 5A). The pointed end structure is supported by 39 crosslinks (Appendix Fig S4C).

The related p25 and p27 both adopt similar left-handed β-helical folds (Yeh *et al*, 2013). They have slight differences in their C-terminal helices, with p27 containing a shorter helix than p25. This, combined with side chain differences and a key crosslink between p25 (residue 175) and p62 (residue 406) (Appendix Figs S9A and B), allowed us to unambiguously assign the two proteins (Fig 5A). p62 adopts an unusual fold (Fig 5B and C, and Appendix Fig S10). The N-terminal and C-terminal β-sheets come together to form a β-sandwich domain. The central portion, which we call the saddle, contains multiple cysteines that fold into three zinc-binding motifs (Fig 5B). There is density between the cysteines in each site (Appendix Fig S9C), which here we model as zincs as this is the most likely occupant (Krishna *et al*, 2003), though another rare possibility is iron (Kluska *et al*, 2018). A long helix extends from the middle of the saddle domain and is followed by a partially disordered loop.

The p62 saddle wraps around the Arp11 subunit at the end of dynactin's filament. The long helix-loop structure folds back across the surface of Arp11 and contacts the neighboring β-actin subunit in

the filament. This interaction is supported by two specific crosslinks between K157 and K222 on p62 to K50 and K61 on actin respectively (Appendix Fig S11A). p25 is located between the p62 β-sandwich and p27. It makes a small contact (134 Å² surface area) with Arp11 (subdomain 2, Appendix Fig S11B) but is predominantly held in place by its interactions with p62 (2746 Å² SA). Its β-helical fold contacts the p62 saddle whereas its C-terminal helix, which is rigidly attached to the β-helical fold, makes interactions with p62's β-sandwich domain. p27 binds to p25 via an extensive interface along their β-helical folds (1,667 Å² SA) (Urnavicius *et al*, 2015). It also makes a small contact with Arp11 (subdomain 4, 263 Å² SA, Appendix Fig S11B) and a number of interactions with p62's saddle (635 Å² SA), albeit fewer than p25.

**Interaction sites for cargo adaptors on the pointed end complex**

Our previous structures showed that dynein's cargo adaptors BICDR1, BICD2, and Hook3 use overlapping, but different sites along the dynactin filament (Urnavicius *et al*, 2015; Urnavicius *et al*, 2018). Here we asked which residues on dynactin's pointed end interact with the different cargo adaptors. We performed signal subtraction on the TDR and TDH datasets individually to focus on the pointed end, which slightly improved the density around the cargo adaptor interaction sites compared with previous maps (Urnavicius *et al*, 2018). We docked our new pointed end structure into these maps and also into our previous dynein tail-dynactin-BICD2 structure (Urnavicius *et al*, 2015).

The majority of pointed end interactions cluster around four sites (Fig 6A and B). Site 1 involves the disordered loop following the long helix in p62 (Fig 6A). It is shared by all three adaptors, with the loop appearing to adopt different conformations to bind each adaptor (Fig EV3A). The other sites are contacted by different subsets of adaptors (Fig 6B). Site 2 is in the p62 saddle region near to p25. Site 3 is in a loop that extends out from p25, whereas site 4 is on the end face of the p25 β-helical fold.

BICDR1 contacts sites 2 and 4. In our previous TDH structure, we noticed two coiled coils at the pointed end (Urnavicius *et al*, 2018). The main one appears to only contact site 3, whereas the second coiled coil binds to site 2, using a different subset of residues compared to BICDR1. It also makes a small contact with a loop in the p62 β-sandwich. BICD2 uses sites 2, 3, and 4. It interacts with another subset of residues at site 2, but the same residues at site 3 and site 4 as used by Hook3 and BICDR1, respectively.

Different coiled-coil cargo adaptors show limited sequence conservation (Reck-Peterson *et al*, 2018). We therefore wanted to ask if their binding sites are conserved. We aligned sequences from a diverse set of eukaryotes that contained the pointed end proteins p62, p25, and p27. In site 2, five of the eight residues strongly conserve their charge (E288, H289, E295, K302, and K304), and two residues (Y32 and F296) are largely aromatic (Fig EV3B). In site 3, p25 residue 74 is always positively charged and residue 76 is often

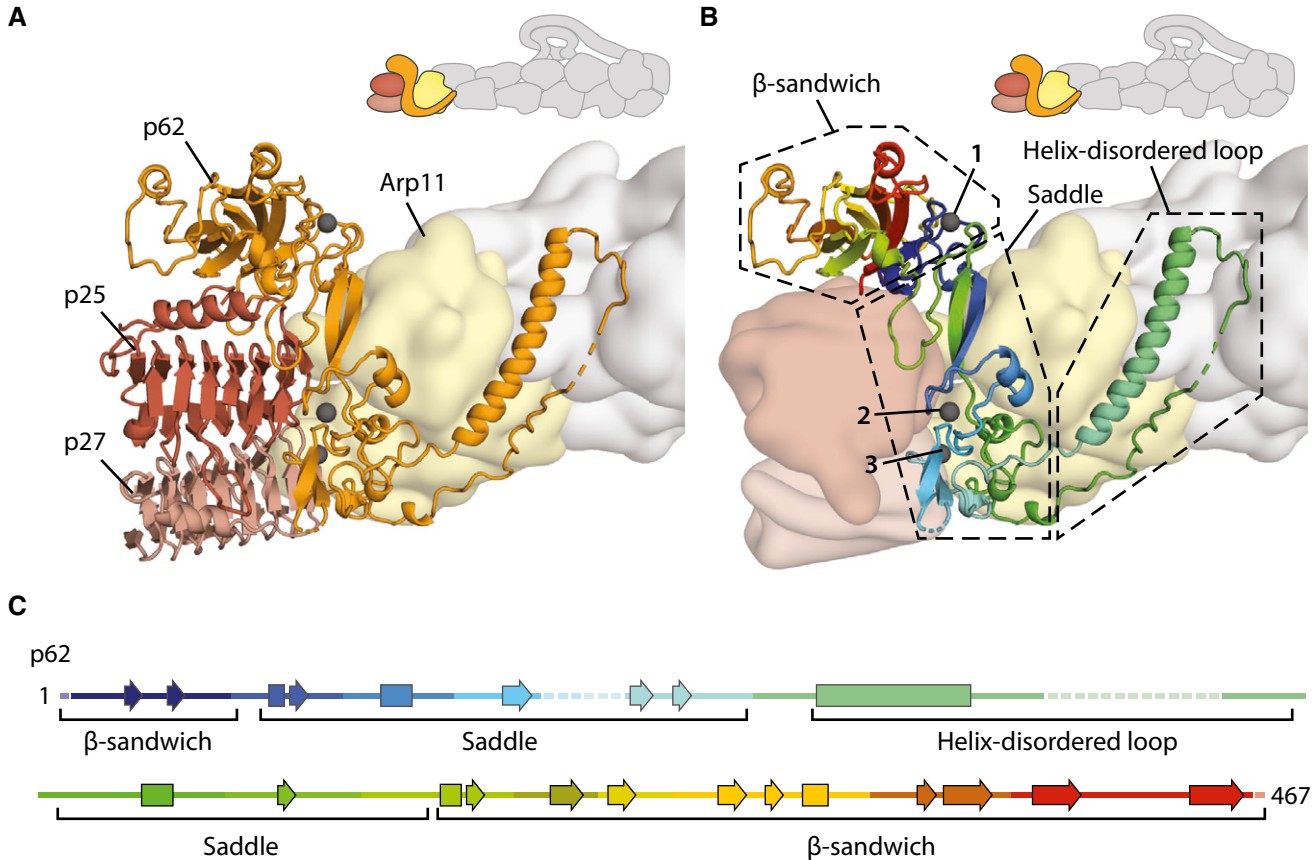

**Figure 5. The organization of the pointed end.**

A  The structure of the pointed end showing p62 (orange), p25 (brown), and p27 (light brown) in cartoon representation. Arp11 (yellow) and the filament are shown as a Gaussian surface rendering of the model. Metal ions in p62 are colored gray, with the metal ion coordinating the N- and C-termini of p62 highlighted (arrow).

B  Detailed structure of p62 (cartoon), colored from N- to C-termini in blue, cyan, green, yellow, orange, and red. Dotted boxes show the different structural features of p62. Metal ions in p62 are colored gray and are numbered.

C  Secondary structure diagram of p62, colored as in (B).

aromatic (Fig EV3C). In site 4, two positions (p25 residues 18 and 31) are strongly conserved as serines or threonines, and residue 32 is conserved as a glutamine (Fig EV3C).

In the case of site 1, residue positions are not well conserved as the loop that contacts the cargo adaptors varies in length. Sequence analysis shows, however, that the first half of the loop maintains a net positive charge (Fig EV4). In our TDR structure, we previously estimated the registry of the BICDR1 coiled coil using density for the sole tryptophan (Trp166) (Urnavicius *et al*, 2018). This positions a series of negatively charged glutamates near the first half of the loop in p62, suggesting an interaction between the cargo adaptor and site 1 at this point.

A plot of the surface conservation of the whole pointed end complex shows that the front side, where the cargo adaptors bind, contains several patches of strong conservation, whereas the reverse face exhibits almost none (Fig 6C). The patches of conservation overlap with sites of adaptor binding. This suggests most adaptors that bind dynactin's pointed end interact with its front face using the sites described here.

To assess the importance of the four adaptor interaction sites, we expressed and purified a pointed end complex consisting of Arp11,

p62, p25, and p27 (Gama *et al*, 2017). We made mutations in sites 1–4 to determine their contribution to cargo adaptor binding. For site 1, we mutated the first half of the disordered loop to a glycine–serine linker. In sites 2–4, we mutated interacting residues to alanine. These mutations did not affect complex composition or stability (Appendix Fig S12A and B). We analyzed the binding of these pointed end complexes to Strep-tagged Hook3 and BICD2 using a pull-down assay (Appendix Fig S12C and D). For both adaptors we saw a large reduction in binding to the site 1 and site 4 mutants, with more minor reductions when site 2 and site 3 were altered (Fig 6D and E). Overall, this mutagenesis together with our structure reveals that sites 1 and 4 are the critical points for adaptor recognition on the pointed end complex.

## Dynactin p150 fold-back sterically blocks all adaptors from binding

In our previous structure of dynactin alone (Urnavicius *et al*, 2015), 10% of the particles showed the p150 arm folded back and docked onto the filament (Fig 7A, cartoon). The region that contacted the pointed end was assigned as two coiled coils from the N terminus of

p150 called CC1A and CC1B (Urnavicius *et al*, 2015; Saito *et al*, 2020).

Here, our cross-linking mass spectrometry data show crosslinks between CC1A, CC1B, and the pointed end of dynactin (Fig 7B, and Appendix Fig S6). This shows that the p150 docked conformation exists in solution and confirms the identity of the two coiled coils.

We also find direct crosslinks from CC1A to CC1B that support their suggested anti-parallel arrangement (Appendix Fig S13) (Tripathy *et al*, 2014; Urnavicius *et al*, 2015; Schroeder & Vale, 2016; Saito *et al*, 2020).

In p150, there is a basic domain and a small globular Cap-Gly domain, N-terminal to the CC1A/B hairpin (Tripathy *et al*, 2014).

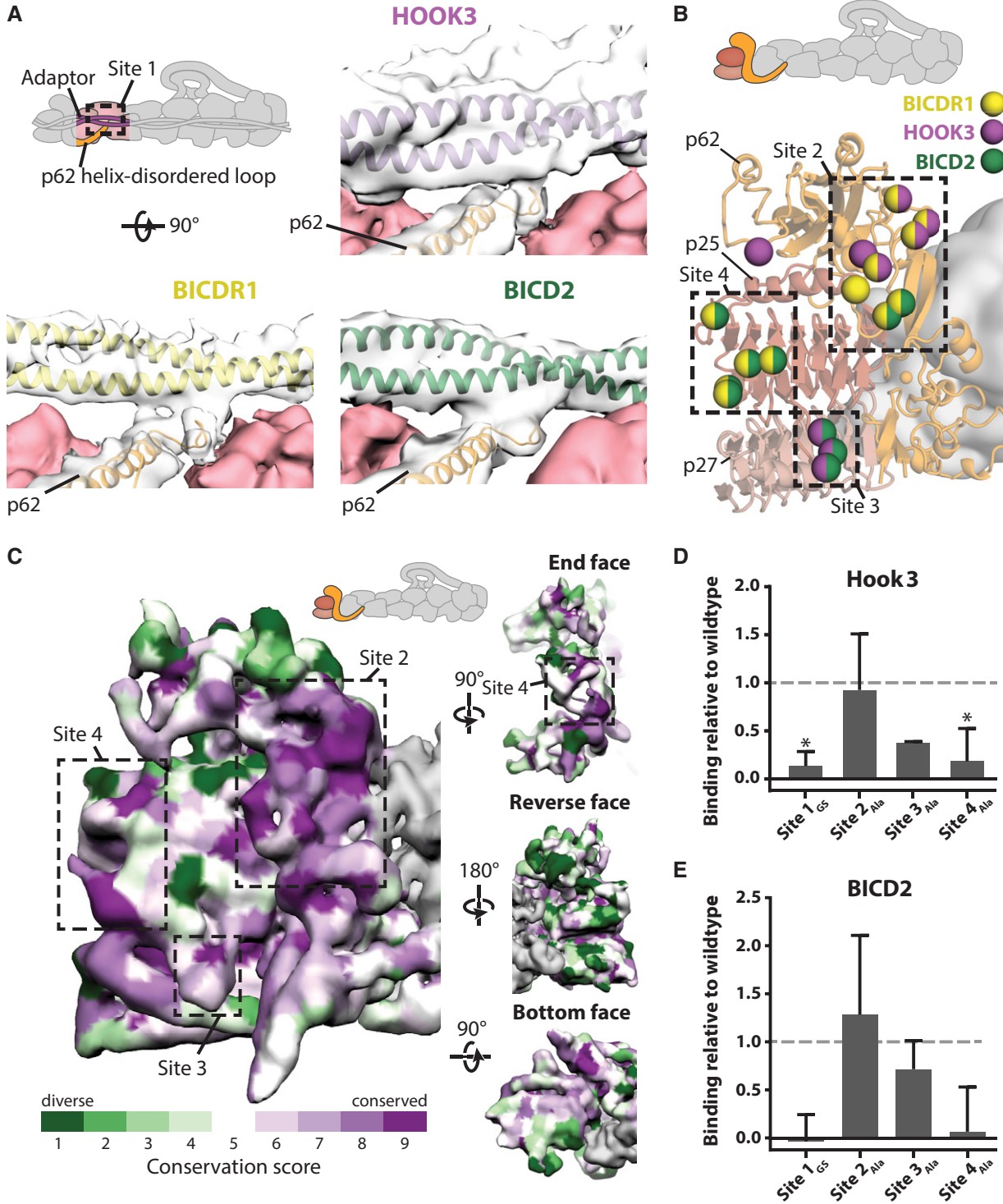

**Figure 6.**

**Figure 6. Pointed end cargo adaptor interaction sites.**

A    Site 1 in the disordered loop of p62 (orange, in clear density), binds to each cargo adaptor (BICDR1 in yellow, Hook3 in purple, BICD2 in green). Density can be seen between p62 and the adaptor at low threshold. BICDR1- and Hook3-containing maps are filtered to 6 Å.

B    Residues on dynactin's pointed end (cartoon) that interact with cargo adaptors are shown as spheres, colored according to which adaptor they bind (yellow denotes binding to BICDR1, purple to Hook3, and green to BICD2). Binding sites 2–4 are shown in boxes.

C    Conservation of the surface residues plotted onto the density from the pointed end map, filtered to 6 Å. Sites 2–4 for cargo adaptor interaction are marked on the front face and end face. Conservation scores were calculated using ConSurf, with lower conservation shown in green, and higher conservation in purple.

D, E    Pull-downs of pointed end site 1–4 mutants using Strep-Hook3 (D) or Strep-BICD2 (E). Binding is shown relative to the binding of the wild-type pointed end construct (dashed lines). * shows $P < 0.05$. For BICD2, $P = 0.06$ for Site 1 mutant, and $P = 0.1$ for Site 4 mutant. Data presented as mean ± SEM, $n = 3$. Statistical significance calculated using ANOVA with Tukey's test for multiple comparisons.

Our cross-linking shows that these regions can contact all parts of the p150 apart from the C-terminal domain, which is buried in the shoulder (Appendix Fig S13). They are also able to interact with Arp11, actin, and the Arp1 filament, consistent with them being highly mobile. In contrast, the tip of the CC1/B hairpin (residues 245–265), which contains lysine residues, makes no crosslinks to other regions. This is consistent with the suggestion that the CC1A/B hairpin, ICD, and CC2 are somewhat rigid and predominantly in an extended conformation (Saito *et al*, 2020).

We attempted to collect more dynactin data to improve the resolution of the docked conformation (Appendix Fig S14). Although this only resulted in a modest improvement in resolution, our new map enabled us to better distinguish the coiled coils and interaction interface. We fit our dynactin structure into the map to examine the residues on the pointed end contacted by the p150 (Fig 7A). This shows that CC1A/B in p150 interacts with distinct set of residues, which overlap with the sites used by cargo adaptors (Fig 7C). CC1A interacts with site 3 and site 4, whereas CC1B covers site 2. This overlap suggests when the p150 arm is bound to the pointed end, all three adaptors (BICD2, BICDR1, and Hook3) will be sterically prevented from binding to dynactin.

## Discussion

### The unique architecture of dynactin's shoulder

Our previous structures showed that the four p50 N-termini emerge from the shoulder and bind to dynactin's filament at four distinct positions (Urnavicius *et al*, 2015; Urnavicius *et al*, 2018). However, it was unclear how the p50 subunits were arranged within the shoulder to facilitate this. Our new structure reveals that each asymmetric half of the shoulder contains two p50 subunits, folded into remarkably different conformations. This arrangement correctly positions each of the four p50 N-termini to bind to its cognate site on the Arp1 filament.

Despite the difference in p50 conformations, the four projecting p50 N-termini are all the same length. Though it had previously been shown that residues 1–87 from p50 could bind the filament (Cheong *et al*, 2014), it became evident from the previous dynactin structures that the sites on the filament to which these termini bind are all different sizes. Here, our structure shows that the four N-termini all exit the shoulder at residue 100. To accommodate the different binding site lengths, the two N-termini bound to the top protofilament are pulled taut. In contrast, the two that bind the bottom protofilament have longer sections of disorder.

We previously observed the helices of p150 entering the shoulder between the arms and splitting off to enter the hook domains. We can now trace the rest of the p150 in the shoulder. After contributing a helical hairpin to the hook domain, the C-terminal 33 residues come together with p50 subunits to form the dimerization domain. This convoluted path allows p150 to interact with both p50 and p24.

In yeast and *C. elegans,* p24 subunits are critical for the incorporation of p150 into the shoulder (Amaro *et al*, 2008; Terasawa *et al*, 2010). Although p24 makes some direct contacts with p150, our work indicates that the main consequence of deleting it would be the incorrect folding of the shoulder because of the extensive p24–p50 interactions. Taken with the previous studies, our structure suggests that all of the interactions that p150 makes in the shoulder are important for its correct incorporation into dynactin.

Because p150 makes an intricate network of interactions with other subunits, it is difficult to imagine it existing in isolation, at least with its current conformation. Previous studies reported the isolated C-terminus of p150 (residues 1050–1286) can interact with potential adaptors including RILP (Johansson *et al*, 2007), SNX6 (Hong *et al*, 2009), and HPS6 (Li *et al*, 2014). It is unclear, however, whether these interactions are possible when the C-terminus is embedded in the shoulder.

### Dynactin's pointed end as an interaction hub

At the pointed end, the structure of p62 was previously elusive. Studies had identified some secondary structural elements (Urnavicius *et al*, 2015) and proposed that a series of cysteines formed zinc-binding motifs (Garces *et al*, 1999; Karki *et al*, 2000). Our structure now resolves p62's β-sandwich, central saddle domain, and long helix regions. Twelve cysteines within the p62 sequence are positioned in three zinc-binding motifs. A previous study predicted that eight of these formed a cross-brace RING domain (Karki *et al*, 2000). We find instead that these residues form two separate folds, both zinc ribbons (as defined by Krishna *et al*, 2003). The final zinc-binding motif uses residues from the N- and C-terminal halves of p62 (Fig 5B, metal ion 1). Similar motifs in other proteins often play structural roles (Coleman, 1992; Lee & Lim, 2008). Hence, these zinc-binding motifs are likely important for the integrity of the elongated structure of p62.

Currently, all the identified cargo adaptors that activate dynein and dynactin for processive movement contain long coiled coils. Previous work reported the structures of three adaptors, BICD2, BICDR1 and Hook3, bound to dynein and dynactin (Urnavicius *et al*, 2015; Urnavicius *et al*, 2018). These structures demonstrated that the three adaptors bind similarly to a few small sites on the

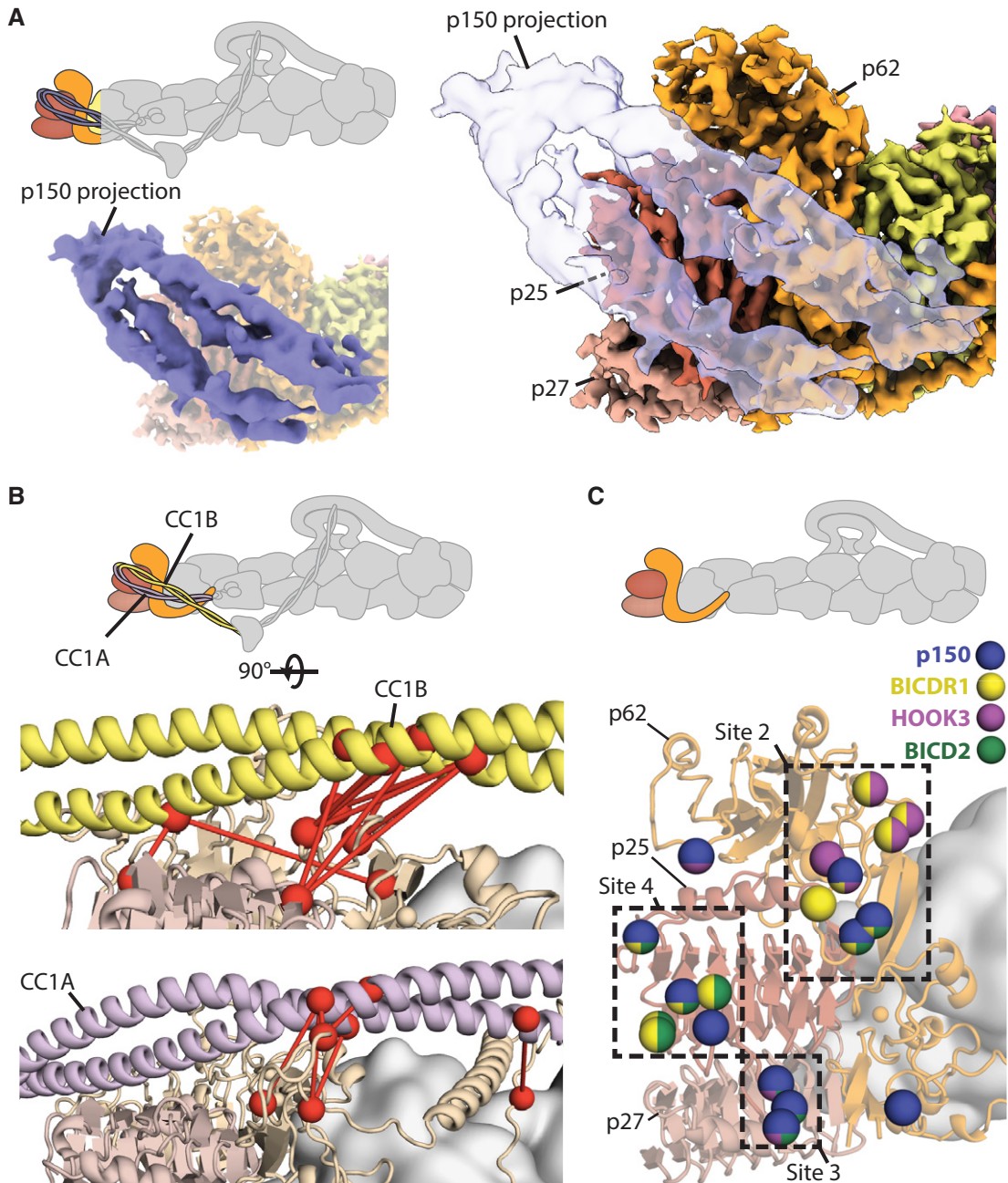

**Figure 7.  Docked p150 interaction sites overlap with adaptor binding sites.**

A   Density representation of the p150-docked structure (solid blue in left inset, transparent in right) superposed over the high-resolution pointed end map shows where the coiled-coil sits.

B   Crosslinks (red dotted lines) connecting residues (shown in red spheres) between either CC1B (upper panel, in yellow) or CC1A (lower panel, in purple) and the pointed end (light orange and light brown).

C   Residues on the pointed end that interact with the p150 or cargo adaptors are shown as spheres. Residues are colored based on their interaction partners. Those that interact with the p150 projection are shown in blue, with interactions with cargo adaptors shown in yellow (BICDR1), purple (Hook3), and green (BICD2). Sites 2–4 are shown in boxes.

filament. However, they lacked the resolution to identify the contact sites on the pointed end (Urnavicius *et al*, 2018). Our work shows that the adaptor-binding residues cluster into four interaction sites. Site 1 and site 2 sit on subunit p62, and site 3 and site 4 are found

on p25. In contrast, we find no sites on p27. This is consistent with its lower surface conservation (Fig 6C), and the observation that p27, unlike p25, is dispensable for some dynactin functions (Gama *et al*, 2017; Qiu *et al*, 2018). Each of the four adaptor-binding sites

consist of a small number of residues and of the four only one, site 3, was predicted from previous modeling data (Zheng, 2017).

Our conservation analysis shows that sites 2, 3, and 4 are well conserved between holozoa (metazoa and closely related single-celled eukaryotes). In fungi and other simple eukaryotes, the sites are conserved only in a subset of species. In fact, many of these species lack a complete pointed end (Hammesfahr & Kollmar, 2012). In *Saccharomyces cerevisiae,* for example, where dynactin is not used for vesicular transport, there appear to be no genes for p62, p25, or p27 (Yeh *et al,* 1995; Moore *et al,* 2009; Hammesfahr & Kollmar, 2012). In contrast, *Aspergillus nidulans*, which contains a coiled-coil cargo adaptor HookA (Zhang *et al,* 2014), has a complete pointed end and conserved sites 2, 3, and 4. Strikingly, however, site 1 is missing in *A. nidulans*, with the whole of the long helix and disordered loop of p62 predicted to be replaced by several shorter helices. Thus, outside of holozoa, the interactions may be different even where coiled-coil adaptors are present.

To investigate the importance of the four adaptor-binding sites, we performed pull-down experiments using an isolated pointed end complex, which has previously been used to investigate cargo adaptor interactions (Gama *et al,* 2017). This shows that mutating sites 1 and 4 reduced adaptor binding for both adaptors we tested (Hook3 and BICD2). In contrast, mutating sites 2 and 3 only had a minor effect. The limited role for sites 2 and 3 is surprising, given their strong sequence conservation (Fig EV3B and C). It raises the possibility that they are important for binding other proteins. This could be other coiled-coil adaptors, or alternatively non-coiled-coil binding partners for dynactin. For example, it has been reported that the non-coiled-coil protein ankyrin-B interacts with the pointed end (Ayalon *et al,* 2011; Lorenzo *et al,* 2014). The importance of site 4 for Hook3 binding was also not expected. Our structures did not show strong density for Hook3 interacting with this part of the pointed end (Urnavicius *et al,* 2018). However, reexamination of our new TDH map at low threshold reveals a weak tube of density contacting site 4 that appears to connect to the main coiled coil of Hook3 via a flexible linker. This mutagenesis, together with the structures of BICD2 and BICDR1 bound to dynein–dynactin, suggests that site 4 is a key interaction site at the pointed end, as discussed below.

### A model for coiled-coil adaptor binding to dynein-dynactin

A number of coiled-coil adaptors for dynein-dynactin have now been identified (Reck-Peterson *et al,* 2018), and a picture is emerging of how they bind (Schroeder & Vale, 2014; Schlager *et al,* 2014b; Gama *et al,* 2017; Reck-Peterson *et al,* 2018). Firstly, they all contact dynein's light intermediate chain (DLIC). The adaptors fall into three families, each of which uses a different DLIC-binding motif/domain: the EF-hand, the Hook domain, and the CC1 box (Lee *et al,* 2018; Lee *et al,* 2020). Secondly, in CC1 box adaptors (e.g., BICD2 and BICDR1), a second motif has been identified, referred to as the CC2 box (Sacristan *et al,* 2018). A third motif, referred to as the Spindly box, has been identified in all three families and evidence suggests that it binds to the pointed end complex (Gama *et al,* 2017). Strangely for BICD2-like adaptors, whereas the distance between the CC1 and CC2 boxes is fixed (Appendix Fig S15), the distance from them to the Spindly box can vary by up to 63 amino acids (between BICD2 and BICDR1). Given the defined

length of dynactin, this raises the question of how the three motifs engage.

Using our registry estimate of BICDR1 (see above, Appendix Fig S15), it becomes clear that the region around the CC2 box interacts with one of the dynein heavy chains (dynein-A2) in the dynein-dynactin-BICDR1 structure (Urnavicius *et al,* 2018). Extending our registry suggests that the positively charged site 1 loop on the pointed end binds to a negatively charged patch on BICDR1 (Appendix Fig S15). The structural flexibility of the site 1 loop (Fig EV3A) indicates that the precise location of its interaction site on each adaptor will differ. In addition, vertebrates contain an alternatively spliced exon in the p62 gene, encoding an additional seven amino acids within site 1 (Fig EV4) (Hammesfahr & Kollmar, 2012). This exon does not contribute to the charge of the loop, but may tune the binding of vertebrate-specific adaptors. The strong effect of mutating site 1 and its conserved positive charge make it likely that patches of negative charge on the adaptor are an important feature for adaptor binding.

Further along the adaptor, our model positions the BICDR1 Spindly box near a prominent kink in the adaptor (Urnavicius *et al,* 2018) and close to site 4 on p25. For BICD2, if the CC1 and CC2 boxes interact in the same way as in BICDR1, we would expect the Spindly box to be too far away to bind the pointed end. However, as previous data suggest the Spindly box is important for BICD2's interaction (Gama *et al,* 2017), the way it interacts with site 4 remains an open question. Although we cannot establish the location of the Hook3 Spindly box in our maps, the observation that Hook3 also depends on site 4 raises the possibility that this site is the major contact point for the Spindly box motif. Intriguingly, one residue in site 4, S31, is both highly conserved (Fig EV3C) and can be phosphorylated (Mertins *et al,* 2014), suggesting this residue may represent a key control mechanism for adaptor recruitment.

As well as binding cargo adaptors, the pointed end has been shown to interact with the p150 projection in previous EM datasets (Urnavicius *et al,* 2015; Saito *et al,* 2020). Our cross-linking mass spectrometry data show that this conformation can occur in solution and our structure shows it would prevent binding of the three cargo adaptors. This suggests that the p150 arm acts to autoinhibit dynactin and that this must be overcome to allow cargo adaptor binding. Autoinhibition is a recurring theme in the molecular motors (Amos, 1989; Verhey & Hammond, 2009; Torisawa *et al,* 2014; Tripathy *et al,* 2014; Terawaki *et al,* 2015; Zhang *et al,* 2017). Dynactin autoinhibition could add another layer of regulation to this complex network. In this model, all three components of the dynein-dynactin-cargo adaptor complex are autoinhibited, with inhibition overcome by stochastic activation and binding of components to drive active complex formation for long processive transport.

# Materials and Methods

### Constructs and sample preparation

The following constructs were used: pACEBac1-HOOK3$^{1-522}$-SNAPf-Psc-2xStrep, pACEBac1-2xStrep-Psc-HOOK3$^{1-522}$ (Urnavicius *et al,* 2018); pACEBac1-Strep-BICD2$^{1-400}$; dynein tail construct containing residues 1–1455 of the human dynein heavy chain (HC) in a pACEBac1

vector with an N-terminal His$_6$-ZZ-TEV tag and fused to pDyn2 (containing genes of human IC2C, LIC2, Tctex1, LC8, and Robl1) as described (Schlager *et al*, 2014a).

We assembled a pointed end complex construct in a pAceBac1 vector comprising human ZZ-TEV-Arp11, p62 (isoform A, UniProt Q9UJW0-1), p25 and p27. To generate mutants in pointed end sites 1–4, we synthesized gene fragments (gBlocks, Integrated DNA Technologies) of the interaction sites, including the required mutations. For site 1, p62 residues 169–195 were mutated to a Gly-Ser linker. For site 2, p62 residues Y32, E281 (equivalent to E288 in *Sus scrofa*), H282 (equivalent to H289), E288 (equivalent to E295), F289 (equivalent to F296), K295 (equivalent to K302), K297 (equivalent to K304), and Q299 (equivalent to Q306) were mutated to alanine. In site 3, p25 residues K74, F76, and K78 were mutated to alanine. In site 4, S18, S31, Q32, V35, and R56 were mutated to alanine. For each case, mutations were made in the vector containing the original subunit, which were then assembled into the pointed end complex construct using Gibson assembly as described previously (Zhang *et al*, 2017).

Dynactin was purified from pig brains using the large scale SP-sepharose protocol (Urnavicius *et al*, 2015). Strep-tagged constructs and the dynein tail construct were expressed and purified using baculovirus as previously described (Urnavicius *et al*, 2018).

The pointed end complex constructs were expressed in *Sf*9 cells (Schlager *et al*, 2014a). For each construct, the frozen cell pellet from 500 ml *Sf*9 cells was thawed in buffer A (50 mM HEPES pH 7.2; 200 mM NaCl; 10% glycerol; 1 mM DTT; 0.1 mM ATP) with a cOmplete™ Protease tablet (Roche) and 1 mM PMSF, total volume 40 ml. When the pellets were thawed, they were lyzed using a Dounce homogenizer. The lysate was clarified at 50,000 *g* for 40 min. Clarified lysates were incubated with 1.5 ml IgG beads (Cytiva), which were pre-equilibrated in buffer A, for 2 h. These beads were then washed using 400 ml buffer A, then 80 ml buffer B (50 mM Tris–HCl pH 7.4; 148 mM potassium acetate; 2 mM magnesium acetate; 1 mM EGTA; 10% glycerol; 1 mM DTT; 0.1 mM ATP). These beads were then transferred to a 2-ml tube, and 100 μl of TEV protease (4 mg/ml) was added, incubated the tube at 4°C for 15 h. Flow through from these beads was collected, concentrated in a 100,000 MWCO Amicon concentrator (Merck), and run on a Superose 6 increase 10/300 column (Cytiva) into buffer C (25 mM HEPES pH 7.2; 150 mM KCl; 1 mM MgCl$_2$; 1 mM DTT; 0.1 mM ATP). These constructs all eluted as a single peak. Each peak was concentrated and stored with 10% glycerol. For each construct, an analytical gel filtration was run of the peak fraction to assess stability in buffer C using a Superose 6 increase 3.2/300 column (Cytiva). Samples were analyzed using NuPAGE Bis-Tris gels (4–12%, 1.0 mm, Invitrogen), staining using InstantBlue Coomassie stain (Expedeon).

For dynein tail-dynactin-Hook3 (TDH) complex grids, dynein tail, dynactin, and HOOK3$^{1-522}$-SNAPf were mixed in a 2:1:20 molar ratio in GF150 buffer (25 mM HEPES pH 7.2; 150 mM KCl; 1 mM MgCl$_2$; 5 mM DTT; 0.1 mM ATP) and incubated on ice for 15 min. The sample was crosslinked to increase the amount of complex formed by addition of 0.0125% (v/v) glutaraldehyde (Sigma-Aldrich), at room temperature for 15 min before quenching with 200 mM Tris pH 7.4 (final concentration). The sample was gel filtered using a TSKgel G4000SW$_{XL}$ (TOSOH Bioscience) equilibrated in 25 mM Hepes-KOH pH 7.2, 150 mM KCl, 1 mM MgCl$_2$, 0.1 mM Mg.ATP,

and 5 mM DTT. The TDH complex was concentrated in a 100 kDa cut-off Amicon centrifugal concentrator (Merck) at 1,500 *g* to 0.1–0.2 mg/ml, and Tween-20 was then added to a concentration of 0.005% (w/v). 3 μl of the TDH sample was applied to freshly glow-discharged Quantifoil R2/2 300-mesh copper grids covered with a thin carbon support. Samples were incubated on grids on a Vitrobot IV (Thermo Fisher Scientific) for 45 s and blotted for 3–4.5 s at 100% humidity and 4°C, then plunged into liquid ethane.

For dynactin grids, dynactin was crosslinked at 350 nM in GF150 buffer using 0.05% glutaraldehyde for 45 min at 4°C. Reactions were quenched using 100 mM Tris pH 7.4. Quantifoil R2/2 300-square-mesh copper grids were covered with a thin carbon support, and glow-discharged for 70 s at 15 mA using a Pelco easiGlow system. 3 μl of sample was then applied to the grids on a Thermo Fisher Vitrobot IV at 100% humidity and 4°C, incubated for 30–40 s, blotted for 3–3.5 s, and then plunged into liquid ethane.

## Cryo-EM data collection and initial data processing for TDH

Electron micrograph movies were recorded using a Titan Krios (Thermo Fisher Scientific) equipped with an energy-filtered K2 detector (Gatan) at 105,000× magnification in EFTEM mode (300 kV, 40 frames, 10 s exposure, ~ 40 e$^-$/Å$^2$). For TDH, movies were acquired in super-resolution mode at the MRC Laboratory of Molecular Biology (0.58 Å/pix), or in counting mode (1.07 Å/pix) at the University of Leeds. Data were collected between 1.5 and 3 μm underfocus using Serial EM or EPU. 4 movies per hole were collected. Correction of inter-frame movement of each pixel and dose-weighting were performed using MotionCor2 (binning the super-resolution data by 2, 5 × 5 patches, excluding first 3 frames) (Zheng *et al*, 2017). CTF parameters were estimated using GCTF (Zhang *et al*, 2016). Micrographs with limited CTF information or with ice contamination were removed at this stage.

For each TDH dataset, Gautomatch (http://www.mrc-lmb.cam.ac.uk/kzhang/) was used to pick particles from all micrographs (4× binned) using 2D classes from EMD-4177 as a reference. 2D classification in RELION 3.0 (spherical mask size 750 Å), combined with manual inspection of particles was used to remove ice, protein aggregates, and other junk particles. Initial 3D refinement was performed using EMD-4177 as an input model, low passed to 60 Å. Each dataset was then cleaned once using 3D classification, using the output from 3D refinement as a reference. At this point, datasets collected at the LMB were merged. These data were then further refined and classified.

After this step, this dataset was merged with data from Leeds and previous data (Urnavicius *et al*, 2018). Pixel sizes were rescaled to match the LMB K2 datasets using Chimera, as described in Wilkinson *et al*, 2019. Briefly, the highest resolution maps from each dataset were compared to the LMB K2 dataset best map in Chimera. The voxel size of the Leeds and previous dataset were then adjusted to maximize the cross-correlation value between the maps from these datasets and the LMB K2 data. This gave us accurate relative pixel sizes, with which we could calculate the scaling factor, as the ratio between the nominal and accurate pixel size for the Leeds and previous dataset. We could use this knowledge to rescale the input particle stacks, and more accurately estimate CTF parameters from their micrographs. A B-factor of +150 Å$^2$ was applied to the K2 dataset, to allow for combination of particles from

different detectors (Zhang *et al*, 2017). After this combination, data were further refined, giving a final reconstruction at 5.6 Å.

### Combination of data

Three datasets were then combined to focus on dynactin: the combined TDH dataset (above); the final particle stack from the tail-dynactin-BICDR1 structure (Urnavicius *et al*, 2018); and the final particle stack from the dynactin structure (Urnavicius *et al*, 2015). Pixel sizes of the TDH datasets were first rescaled to match the TDR dataset (1.34 Å/pix), as described above. For TDR and TDH datasets, density for dynein and adaptors were then subtracted in two steps, each using a 4 pixel soft-edge mask (Fig EV1, step A). Dynein heavy chains (from residue 467 to its C-terminus) and dynein light intermediate chains were first subtracted in RELION 3.0. The output particles were refined to more accurately align the remaining density. We then used a second round of signal subtraction to remove the remainder of the dynein signal and the cargo adaptor. After these steps, data could be combined (Fig EV1, step B). The combined dataset was then subjected to a round of global refinement, initially using a 600 Å spherical mask, then a 6 pixel soft-edge mask. This resulted in an overall dynactin reconstruction (EMD-11313) at a resolution of 3.8 Å.

### Data processing for the shoulder and pointed end

Processing for the dynactin shoulder was performed in RELION 3.0. Different masks were tried with signal subtraction to improve the density of the shoulder (Fig EV1, step C). All of the masks tried were created using a low-pass filter of 15 Å and 6 pixel soft edge. Masks of the shoulder components alone or including a small portion of the underlying filament did not have enough signal to align. In contrast, a mask including shoulder components and five underlying filament subunits improved the density of the shoulder. This mask was optimized using focused refinement without signal subtraction (Fig EV1, step D, Appendix Fig S1). The map from this refinement was closely examined to see ordered density outside the mask and blurred density inside the mask. Using this optimization technique, we found that the Arp1 subunit A nearest the barbed end could be removed from the mask, while half of Arp1 subunits F and G should be included. Using local resolution on the same map identifies similar regions to optimize. Signal subtraction was then performed, subtracting the signal outside of this mask. This was followed by local refinement using a 6 pixel soft-edge mask (Fig EV1, step E). The CTF parameters were then refined. This was succeeded by a round of 3D classification without alignments (Fig EV1, step F). Due to the small size of the particle in comparison with the box size, a T-value of 50 was used, with 25 classes. The best class was then locally refined using a 6 pixel soft-edge mask. This map, EMD-11314 (Fig EV1, step G), was at an overall resolution of 3.8 Å and was used to build the majority of the shoulder. After CTF refinement, signal subtraction also was used to focus on a map consisting of the upper paddle, upper hook, and distal region of the lower arm, using a 4 pixel soft-edge mask (Fig EV1, step H). After signal subtraction, 3D classification was performed without alignments using a T-value of 180, 16 classes, and limiting resolution to 6 Å in the expectation step. We chose the class from this containing the most ordered density and reverted to non-subtracted shoulder particles, subjecting these particles to a local masked refinement. This resulted in a map

with an overall resolution of 4.6 Å. This map, EMD-11316 (Fig EV1, step I), was low passed to 6 Å for modeling the upper paddle, upper hook, and distal region of the lower arm.

Different masks were also tried with signal subtraction to improve the density of the pointed end (J). All of the masks tried were created using a low-pass filter of 15 Å and 6 pixel soft edge. For the signal subtraction using the best mask, RELION 3.1 was used (Zivanov *et al*, 2020). This version of RELION allowed us to recenter particles on mask's center-of-mass during subtraction and also to reduce the box size. This recentering feature was important to improve the resolution of the pointed end. This method compared favorably with attempts at multi-body refinement in RELION, which does not recenter particles as part of its workflow. Subtraction was accomplished in two stages to ensure accurate subtraction. First, half of the dynactin, including the barbed end and shoulder subunits, was subtracted using a 6 pixel soft-edge mask (Fig EV1, step K). The output was locally refined using a 6 pixel soft-edge mask, to enable further subtraction using more accurate angles for the remaining half of dynactin. In our first attempt, this subtraction resulted in striated artefactual density, near the β-sandwich domain of p62 (Fig EV1, marker 1). By closely examining previous maps, we determined that some signal from the β-sandwich domain of p62 had been removed when subtracting the density for the adaptor in the TDR and TDH complexes. We hence went back to these complexes and repeated the signal subtraction of the adaptors using a tighter mask around the adaptors to prevent subtraction of p62 signal. After processing this new subtraction as above, the β-sandwich domain of p62 was better resolved (Fig EV1, marker 2). We then optimized the mask, to exclude more of the filament in a subsequent second round of subtraction (Fig EV1, step L). We subjected these particles to 3D classification with no alignments, performed with a T-value of 50, 25 classes (Fig EV1, step M). The best class was then locally refined using a 6 pixel soft-edge mask to an overall resolution of 4.1 Å (EMD-11315) (Fig EV1, step N).

To examine the pointed end for TDR and TDH, the complex datasets were kept separate after the first step of dynein subtraction, rather than combining these data as above. For each complex, a mask around the pointed end was created including the adaptor. Signal outside of this mask was subtracted using RELION 3.1 to leave the pointed end with adaptor attached. These maps were then subjected to a round of local refinement.

All maps were post-processed using RELION 3.1, with B-factors initially estimated automatically (Rosenthal & Henderson, 2003). For building, each map was processed using multiple B-factors between the estimated value and zero and low passed to different resolutions, to account for the heterogeneity in resolution in the maps. For examining the interface with adaptors, lower B-factors were used, as at high B-factors signal for the adaptors was lost. Local resolution was calculated in RELION 3.1 (Kucukelbir *et al*, 2014). EMDA (Warshamanage & Murshudov, https://www2.mrc-lmb.cam.ac.uk/groups/murshud ov/content/emda/emda.html) was used to align shoulder and pointed end maps to the map of the whole dynactin. This avoids the information loss inherent in the resampling procedure in Chimera.

### Cryo-EM data collection and data processing to study dynactin p150 docked conformation

Electron micrograph movies were recorded using a Titan Krios (Thermo Fisher Scientific) equipped with an energy-filtered K2

detector (Gatan) in at 105,000× magnification in EFTEM mode (300 kV, 40 frames, 10 s exposure, ~ 40 e$^-$/Å$^2$). Movies were acquired in counting mode for dynactin (1.16 Å/pix). Data were collected between 1.5 and 3 μm underfocus using Serial EM or EPU. 4 images per hole were collected. Correction of inter-frame movement of each pixel and dose-weighting were performed using MotionCor2 (5 × 5 patches) (Zivanov *et al*, 2018). CTF parameters were estimated using GCTF. Micrographs with limited CTF information or with ice contamination were removed at this stage.

For dynactin datasets, Gautomatch (http://www.mrc-lmb.cam.ac.uk/kzhang/) was used to pick particles from all micrographs (4× binned) using 2D projections of EMD-2856 as a reference. 2D classification in RELION, combined with manual inspection of particles was used to remove ice, protein aggregates, and other junk particles. Initial 3D refinement was performed using EMD-2856 as an input model, low passed to 60 Å. Each dataset was then cleaned once using 3D classification, using the output from 3D refinement as a reference. At this point, datasets were unbinned and merged. This combined dataset was aligned using 2D classification and then subjected to 3D refinement, classification, and Bayesian polishing. The final reconstruction showed more flexibility in the shoulder and pointed end than our complex datasets. As a result, combining these data with the combined dataset did not improve the resolution of the shoulder of the pointed end maps.

To focus on the p150 docked conformation, these data were then merged with the particles from our combined dataset. Signal subtraction was used to focus on the pointed end, using a loose mask to ensure the p150 density was not subtracted. 3D classification without alignments was then used (16 classes, T20) to separate out dynactin particles containing a docked p150 arm, using a loose mask to include density for the p150. The best class from this was then locally refined to a global resolution of 6.8 Å.

**Model building and refinement**

Building was performed in COOT (Emsley & Cowtan, 2004; Emsley *et al*, 2010). For the shoulder, we first used real-space refine in COOT to refine the dynactin model from the previous TDR structure, PDB 6F1T (Urnavicius *et al*, 2018), into our new density. All secondary structure elements were first rebuilt if necessary, and fit into density.

Sidechains could be built in many regions, allowing us to unambiguously assign much of the shoulder. In the lower subdomain, we could build sidechains in the hook, paddle, dimerization domain and in part of the arm region. We built sidechains for p50-A (PDB chain m) in the following regions: residues 82–186; 207–215; 224–245; 261–273; and 310–364. For p50-B (chain n), we could build sidechains for 98–122; 137–183; 208–244; 261–273; 314–364. For p24 (chain o), we could build side chains for 7–41; 83–137. For p150 (chain Z), we could build sidechains for 1096–1143, 1155–1184, 1188–1258, 1266–1285.

In the upper subdomain, we had sidechain density for the dimerization domain and the majority of the arm region. We built sidechains for p50-A (PDB chain M) for residues 175–183; 210–214; 312–400. In p50-B (chain N), we can build sidechains for residues 210–217; 310–402. For p24, we can build sidechains for (chain O) 79–176. In p150 (chain z), we can build sidechains for residues 1096–1124; 1253–1258; 1266–1285.

Starting from these sidechain-resolved regions, we could trace and assign much of the rest of the shoulder. For lower tetramer p50-A (PDB chain m) residues 82–187 and 207–399 can be traced, with smeared density apparent at low threshold we assign as the loop consisting of residues 187–207. The lower p50-B (chain n) can be traced from residue 98–187, 207–244, and 260–393, with clear density at low threshold for the loop comprising residues 187–207. p24 (chain o) can be traced from residue 4–173. The p150 (chain Z) from the lower tetramer can be traced from 1092 to 1285.

In the upper tetramer, density was at lower resolution in the hook and paddle, which house the N-terminal halves of the p50s, and residues 1159–1220 of p150. We used the equivalent region in the lower tetramer to assist in assigning the secondary structure elements. We could trace p50-A (chain M) residues 102–188, 209–243, 261–403; p50-B (chain N) residues 98–186, 209–242, 260–403; p150 residues 1092–1178, 1180–1219, 1252–1258, 1266–1285. In addition, we could trace p24 (chain O) residues 2–180.

Once we had built the shoulder, we could link the p50s to the filament-binding N-termini that had been previously built (Urnavicius *et al*, 2015). We could build the connection between p50-A (chain m) from the lower subdomain to the adjacent p50 N terminus. We could see density at lower threshold to connect p50-B from the lower subdomain (chain n) to the N terminus on the opposite side of the filament. For the upper subdomain, we tentatively assign the N-termini of p50-A (chain M) and p50-B (chain N), based on their geometry in the paddle.

For the pointed end, we used real-space refine in COOT to refine the pointed end subunits from the previous TDR model (PDB 6F1T) into our density. We could then assign and build side chains for p25 (residues 3–176) and p27 (residues 9–174). For p62, side chains for its long helix were first modeled. We then built the rest of the structure, assigning side chains in the saddle region, and part of the β-sandwich: residues 2–87; 107–167; 221–321; 332–337; 368–374; 401–417; 422–430; 453–463. Zincs were modeled into the three zinc-binding motifs based on the coordinating ligands. Two loops were not modeled due to lack of density, indicating extreme flexibility. The first (residues 89–105) is between two adjacent β-strands. The second (residues 183–218) was the disordered loop following the long helix in p62.

Model refinements were performed in REFMAC5 (Murshudov *et al*, 1997; Nicholls *et al*, 2018). The entire dynactin model was first refined into the overall dynactin map to best fit the model into the density. For further refinement, the model was then split into three sub-models corresponding to each of the two rigid bodies used for masked refinement of the shoulder and pointed end, and the remainder of the filament. These filament subunits were refined into the overall map. Refinement proceeded on these three sub-models separately, iterating between manual optimization of model geometry using coot "real-space refine" and automated real-space refinement in REFMAC5. These refinements consisted of 20 iterations using a refinement weighting of 0.0001, with hydrogen atoms included. The early refinements on the filament imposed non-crystallographic symmetry onto the 7 barbed-end proximal Arp1 subunits (Chain ID A-G). In the first refinements of both the pointed end and the shoulder, the maps were initially low-pass filtered to 6 Å in order to fit the model into the areas with lower resolution. The high-resolution maps were then used for subsequent refinements, in which the side chain conformations were optimized. For each

sub-model, refinements continued until the model validation scores stopped improving, as calculated using PHENIX validation tools (Afonine *et al,* 2018; Williams *et al,* 2018). Boundaries between sub-models were refined in PHENIX, using restraints on other parts of the model.

To assess pointed end interactions with cargo adaptors/p150 arm, we first used rigid-body fitting to place our new pointed end model into the appropriate map: EMD-11317 for pointed end-BICDR1 interactions; EMD-11318 for pointed end-Hook3 interactions; EMD-2860 for pointed end-BICD2 interactions (Urnavicius *et al,* 2015); and EMD-11319 for pointed end-p150 interactions. We used rigid-body fitting to place previous structures of TDB (PDB 6F3A), TDR (PDB 6F1T), TDH (PDB 6F38), or dynactin with the p150 (PDB 5ADX) into the appropriate map and then removed components other than the cargo adaptor/p150 arm. For the pointed end-BICDR1 and pointed-BICD2 models, cargo adaptors were refined using PHENIX (at 6 Å and 8 Å, respectively), to resolve clashes with the pointed end. For the pointed end-Hook3 model, the second coiled coil was modeled into density. For the p150 arm, the registry of the coiled coils was updated to be consistent with our mass spectrometry cross-linking data. We then examined the density connecting our new pointed end structure with the cargo adaptor/p150 arm to discern likely interacting residues. Contact sites were only considered where there is density for both dynactin's pointed end and cargo adaptor, with connecting density. Solvent accessible surface areas were approximated using Pymol.

**Cross-linking mass spectrometry**

200 μg dynactin at 3 μM in GF150 buffer was crosslinked with 1.5 mM BS3 for 2 h at 4°C. The reaction was then quenched using 160 mM Tris pH 7.4. The crosslinked samples were cold-acetone precipitated and resuspended in 8 M urea and 100 mM $NH_4HCO_3$. Proteins were reduced with 10 mM DTT and alkylated with 50 mM iodoacetamide. Following alkylation, proteins were digested with Lys-C (Pierce) at an enzyme-to-substrate ratio of 1:100 for 4 h at 22°C and, after diluting the urea to 1.5 M with 100 mM $NH_4HCO_3$ solution, and further digested with trypsin (Pierce) at an enzyme-to-substrate ratio of 1:20.

Digested peptides were eluted from StageTips and split into two for parallel crosslink enrichment by strong cation exchange chromatography (SCX) and size exclusion chromatography (SEC) and were dried in a vacuum concentrator (Eppendorf). For SCX, eluted peptides were dissolved in mobile phase A (30% acetonitrile (v/v), 10 mM $KH_2PO_4$, pH 3) before strong cation exchange chromatography (100 × 2.1 mm PolySulfoethyl A column; Poly LC). The separation of the digest used a gradient into mobile phase B (30% acetonitrile (v/v), 10 mM KH2PO4, pH 3, 1 M KCl) at a flow rate of 200 μl/min. Ten 1-min fractions in the high-salt range were collected and cleaned by StageTips, eluted, and dried for subsequent liquid chromatography with tandem mass spectrometry (LC-MS/MS) analysis. For peptideSEC, peptides were fractionated on an ÄKTA Pure system (GE Healthcare) using a Superdex Peptide 3.2/300 (GE Healthcare) at a flow rate of 10 μl/min using 30% (v/v) acetonitrile and 0.1% (v/v) trifluoroacetic acid as mobile phase. Five 50-μl fractions were collected and dried for subsequent LC-MS/MS analysis.

Samples for analysis were resuspended in 0.1% v/v formic acid, 1.6% v/v acetonitrile. LC-MS/MS analysis was conducted in

duplicate for SEC fractions and triplicate for SCX fractions and performed on an Orbitrap Fusion Lumos Tribrid mass spectrometer (Thermo Fisher Scientific) coupled online with an Ultimate 3000 RSLCnano system (Dionex, Thermo Fisher Scientific). The sample was separated and ionized by a 50 cm EASY-Spray column (Thermo Fisher Scientific). Mobile phase A consisted of 0.1% (v/v) formic acid and mobile phase B of 80% v/v acetonitrile with 0.1% v/v formic acid. Flow rate of 0.3 μl/min using gradients optimized for each chromatographic fraction from offline fractionation ranging from 2% mobile phase B to 45% mobile phase B over 90 min, followed by a linear increase to 55% and 95% mobile phase B in 2.5 min, respectively. The MS data were acquired in data-dependent mode using the top-speed setting with a three second cycle time. For every cycle, the full scan mass spectrum was recorded in the Orbitrap at a resolution of 120,000 in the range of 400–1,600 *m/z*. Ions with a precursor charge state between 3+ and 7+ were isolated and fragmented. Fragmentation by higher-energy collisional dissociation (HCD) employed a decision tree logic with optimized collision energies (Kolbowski *et al,* 2017). The fragmentation spectra were then recorded in the Orbitrap with a resolution of 30,000. Dynamic exclusion was enabled with single repeat count and $60^{-s}$ exclusion duration.

A recalibration of the precursor *m/z* was conducted based on high-confidence (< 1% false discovery rate (FDR)) linear peptide identifications. The recalibrated peak lists were searched against the sequences and the reversed sequences (as decoys) of crosslinked peptides using the Xi software suite (v.1.6.745) for identification (Mendes *et al,* 2019). The following parameters were applied for the search: MS1 accuracy = 3 ppm; MS2 accuracy = 10 ppm; enzyme = trypsin (with full tryptic specificity) allowing up to four missed cleavages; crosslinker = BS3 with an assumed reaction specificity for lysine, serine, threonine, tyrosine, and protein N-termini; fixed modifications = carbamidomethylation on cysteine; variable modifications = oxidation on methionine, hydrolyzed/aminolyzed BS3 from reaction with ammonia or water on a free crosslinker end. The identified candidates were filtered to 2% FDR on link level using XiFDR v.1.1.26.58 (Fischer & Rappsilber, 2017).

Crosslinks were then plotted on the structure in Pymol, and their lengths were calculated. Disordered loops were modeled in Coot, to assess instances where at least one crosslinked residue was in a loop. The p150 projection, the flexible N- and C-termini of Arp11, and the disordered loop in p62 were not modeled, due to their length. Crosslinks under 30 Å ($C_\alpha$-$C_\alpha$ distance) were considered valid. Crosslinked residue pairs over 30 Å apart were examined to see if small changes in the conformation of dynactin could enable valid crosslinks to form. For crosslinks between subunits where multiple copies exist in dynactin (e.g., Arp1), the shortest crosslink was assessed.

**Bioinformatics**

PSI-BLAST and JACKHMMER profile-based sequence searches were used to identity eukaryotic homologs of p25, p27, and p62. Toward this, we provided the sequences of p25, p27, and p62 from *Sus scrofa* as initial query inputs to search against UniProt/TrEMBL databases (with an e-value cut-off = 0.001). Further, more distant homologs of p25, p27, and p62 of *Sus scrofa* were identified by providing respective multiple sequence alignments of the first set of

above-identified homologs as queries to JACKHMMER to search against UniProt/TrEMBL (e-value = 0.001). Sequences from a diverse set of eukaryotes were aligned using MAFFT (Madeira *et al*, 2019). Weblogo was used to visualize these alignments for the interacting residues (Crooks *et al*, 2004). ConSurf was used to calculate per-residue conservation scores from these alignments (Ashkenazy *et al*, 2016). To visualize surface conservation, conservation scores were rendered on the models of the pointed end. Sidechain conservation was then rendered onto the density for dynactin's pointed end. The standard ConSurf color-blind-friendly color scheme was used for visualization.

### Pull-down assays

For the pull-down assays, 100 pmol of Strep-tagged adaptor was incubated with 700 pmol of the pointed end complex in 40 μl (final volume) of buffer D (25 mM HEPES pH 7.2; 100 mM KCl; 1 mM $MgCl_2$; 1 mM DTT), for 1 h at 4°C. Each sample was then diluted to 150 μl with buffer D, the incubated with pre-equilibrated 30 μl Streptactin beads (IBA Lifesciences) for 20 min, mixing regularly. Beads were then spun down at 20 *g*, 3 min and then washed using buffer E (25 mM HEPES pH 7.2; 150 mM KCl; 1 mM $MgCl_2$; 1 mM DTT), 5 × 500 μl. Beads were incubated with 100 μl elution buffer (buffer E plus 3 mM desthiobiotin) for 10 min and then spun down at 20 *g*, 3 min. This pull-down was completed in triplicate, using a different preparation of each pointed end construct in every replicate.

Eluant samples were run on NuPAGE 4–12% Bis-Tris SDS–PAGE gel (Thermo Fisher Scientific) and stained using SYPRO Ruby gel stain (Thermo Fisher Scientific). Gel images were acquired using a Gel Doc XR + Imaging System (Bio-Rad). For each replicate, all samples were run on the same gel. The amount of pointed end complex was assessed via its Arp11 band, as quantified in ImageJ. For each construct, the intensity in the sample without the Strep-tagged adaptor was subtracted from the samples including Strep-Hook3 or Strep-BICD2. Then, the amount of binding of each mutant was calculated, relative to wild-type binding for that replicate.

Statistics were generated using GraphPad Prism 7. For each adaptor, an ANOVA was carried out, correcting for multiple comparisons using Tukey's test.

### Structure rendering for figures

Density images for figures were rendered using ChimeraX (Goddard *et al*, 2018) or Chimera v1.13.1 (Pettersen *et al*, 2004), and model-only images were rendered using Pymol (DeLano, 2002). Pymol was used to render surface representations of the model, using a Gaussian isosurface.

## Data availability

The data produced in this study are available in the following databases:

- Cryo-EM maps: EMDB
  a. EMD-11313 (https://www.ebi.ac.uk/pdbe/entry/emdb/EMD-11313)
  b. EMD-11314 (https://www.ebi.ac.uk/pdbe/entry/emdb/EMD-11314)
  c. EMD-11315 (https://www.ebi.ac.uk/pdbe/entry/emdb/EMD-11315)
  d. EMD-11316 (https://www.ebi.ac.uk/pdbe/entry/emdb/EMD-11316)
  e. EMD-11317 (https://www.ebi.ac.uk/pdbe/entry/emdb/EMD-11317)
  f. EMD-11318 (https://www.ebi.ac.uk/pdbe/entry/emdb/EMD-11318)
  g. EMD-11319 (https://www.ebi.ac.uk/pdbe/entry/emdb/EMD-11319)
- Model coordinates: PDB
  a. 6ZNL (https://www.ebi.ac.uk/pdbe/entry/pdb/6znl )
  b. 6ZNM (https://www.ebi.ac.uk/pdbe/entry/pdb/6znm)
  c. 6ZNN (https://www.ebi.ac.uk/pdbe/entry/pdb/6znn)
  d. 6ZNO (https://www.ebi.ac.uk/pdbe/entry/pdb/6zno)
  e. 6ZO4 (https://www.ebi.ac.uk/pdbe/entry/pdb/6zo4 )
- Crosslinking mass spectrometry data: PRIDE PXD020084 (http://www.ebi.ac.uk/pride/archive/projects/PXD020084)

Expanded View for this article is available online.

## Acknowledgements

We thank Linas Urnavicius and Kai Zhang for helpful discussion and datasets for TDR and dynactin; Takanori Nakane, Sjors Scheres, Garib Murshudov, Alexey Murzin, and Shabih Shakeel for helpful discussions; Jake Grimmett and Toby Darling for scientific computing support; and Ferdos Abid Ali and Sami Chaaban for comments on the manuscript. We acknowledge the MRC—Laboratory of Molecular Biology Electron Microscopy Facility for access and support of electron microscopy sample preparation and data collection; the University of Leeds Electron Microscopy facility for help with data collection and Diamond for access and support of the Cryo-EM facilities at the UK national electron bio-imaging center (eBIC), proposal EM18086, funded by the Wellcome Trust, MRC and BBSRC. This work was funded by grants from the Wellcome Trust (WT210711) and the Medical Research Council, UK (MC_UP_A025_1011) to APC, Wellcome Trust Senior Research Fellowship (103139) to JR, Deutsche Forschungsgemeinschaft project no. 426290502.

## Author contributions

CKL and APC conceived the research. CKL performed the cryo-EM work, cloned, expressed and purified proteins, and performed pull-down assays. FJO and JR conducted the cross-linking mass spectrometry experiments. BS collated and aligned the pointed end protein sequences. CKL and SEL built and refined the dynactin model. APC and CL prepared the manuscript. All authors edited the manuscript.

## Conflict of interest

The authors declare that they have no conflict of interest.

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
