## [Review Process File · The EMBO Journal]

Cryo-EM reveals the complex architecture of dynactin's shoulder region and pointed end

Clinton Lau, Francis O'Reilly, Balaji Santhanam, Samuel Lacey, Juri Rappsilber, and Andrew Carter DOI: [10.15252/embj.2020106164](https://doi.org/10.15252/embj.2020106164)

Corresponding author: Andrew Carter (cartera@mrc-lmb.cam.ac.uk)

Review Timeline:

Submission Date:	7th Jul 20
Editorial Decision:	24th Aug 20
Revision Received:	18th Dec 20
Editorial Decision:	5th Feb 21
Revision Received:	11th Feb 21
Accepted:	17th Feb 21

Editor: Ieva Gailite

Transaction Report:

Thank you for submitting your manuscript for consideration by The EMBO Journal. I apologise for the protracted review process of your manuscript caused by delays in review submission. We have now received three referee reports on your manuscript, which are included below for your information.

As you will see from the comments, the reviewers appreciate the study, but they also find that the presentation and characterisation of the structural data should be improved before they can recommend acceptance of the manuscript. Furthermore, reviewer #2 finds that further functional investigation of the identified cargo binding sites and p62 structural motifs would be important. From my side, I find the reviewer comments generally reasonable. Therefore, I would invite you to address the concerns raised by the reviewers in a revised manuscript.

I should add that it is The EMBO Journal policy to allow only a single major round of revision and that it is therefore important to resolve the main concerns at this stage. We are aware that many laboratories cannot function at full efficiency during the current COVID-19/SARS-CoV-2 pandemic, and I would be happy to discuss the revision in more detail via email or phone/videoconferencing.

Please feel free to contact me if you have any further questions regarding the revision. Thank you for the opportunity to consider your work for publication. I look forward to receiving the revised manuscript.

Referee #1:

Cryo-EM reveals the complex architecture of dynein's shoulder and pointed end
Dynein is responsible for most of the minus-end directed transport along microtubules and is capable of transporting many different cargoes. To become processive, dynein must be activated by binding an activating adaptor protein and the multi-subunit protein complex dynactin. The structure of dynactin was determined to near atomic resolution several years ago, which allowed mapping of how dynein and adaptor proteins form a tripartite complex with dynactin. However, due to the flexible nature of the complex, several areas of dynactin were not resolved to high resolution, including the pointed end and shoulder regions, both of which are structurally and functionally important.

In this manuscript, Lau et al combine particles from prior cryo-EM datasets and reprocess the data to improve the resolution of the pointed end and shoulder domains. The authors provide a detailed description for their rationale behind choosing which masks are used for particle signal subtraction and local refinement. The higher resolution maps allow the authors to assign previously unknown densities to specific subunits of both the shoulder and pointed end. Subunit assignment was validated using crosslinking and mass spectrometry. Finally, atomic models were built for the

shoulder and pointed end using their high-resolution maps. These models allowed the authors to show different adaptor proteins have distinct binding sites on the pointed end. Combining a modest increase in resolution for the docked conformation of p150 and their cross-linking data, the authors also propose that binding of p150 in that conformation would sterically compete with all three of the activating adaptors characterized to date (BICD2, BICDR1 and Hook3), adding another layer of regulation to an already baroque complex molecular motor.

This is a well-written, straight-forward description of a significant improvement in major components of dynactin, a central regulator of microtubule-based transport. This is an important advance of interest to a broad readership. The description of the data processing also makes the manuscript interesting to anyone in the cryo-EM community working on similarly complex systems.

There are no major problems with the manuscript. All the comments below are minor suggestions I believe could help improve it.

Major issues

- None

Minor issues / Comments

- The authors describe sequence conservation between the adaptor proteins and interaction sites on the pointed end. Did they look to see if these sites show any evolution covariance? That might be more informative than looking at general sequence conservation of the individual proteins.
- The authors performed CTF refinement in RELION 3.0. It would be interesting to see if the resolution improved after CTF refinement in RELION 3.1 using the higher order aberration and anisotropic magnification refinement options.
- The new p150 docked data should be included in the Supplementary 1 processing figure. Why wasn't the new data included with the rest of the processing if particle count was a concern?
- How does particle subtraction compared with doing multibody refinement?
- Figure 2A: It is slightly confusing that a strongly filtered map seems to be shown in this panel rather than the one that would have been used to build the model. Given that a major point of this paper is the improvement of the density, it would be less confusing to readers if a higher resolution map were shown here.
- Figure 2B: I found the choice of colors confusing here as well. Blue, yellow and red are used in panel (A) to distinguish the Lower arm, Upper arm and dimerization domain. In panel (B), red and yellow are used again but this time to distinguish the proteins comprising the Lower arm, and the purple used for p150 is fairly close to the blue used in (A).
- Figure 2E: It is not possible to tell the handedness of the rotation from the symbol without introducing some breaks in the arrow/segment.
- Figure 3A/B: As with Figure 2, I found it confusing that the authors use a strongly filtered version of their maps rather than the one used for model building. Is the problem that the level of detail in the higher-resolution map distracts from the ribbon diagrams? Would a more transparent setting help with that? If the authors feel this is the best way of presenting things, they may want to add a line to the figure legend (here and in Figure 2) explaining that the surface representation has been filtered for the purposes of the figure.
- Figure 4E: The color choices in this panel are also potentially confusing as green and purple are used for amino acids in panels (C) and (D).
- Figure 5C: It would be nice to include in this panel the same dashed boxes shown in Figure 4B for binding sites 2-4 (this would prevent the reader from having to go back and forth between them).

Very minor syntax point

- Line 27: "Dynactin is a 1.1 MDa complex that activates the molecular motor, dynein, for ultra-processive..." and Line 42: Dynactin is a large, multi-subunit co-activator for the molecular motor, cytoplasmic dynein 1." It seems to me that the commas in "...dynein,..." and "..., cytoplasmic dynein 1" should not be there. "Dynein" and "cytoplasmic dynein 1" are not clauses that can be removed from those sentences without changing their meaning, as dynein is not the only molecular motor. Only "dynein" or "the molecular motor dynein" would uniquely identify it.

Referee #2:

Summary:

In this work, the authors take advantage of three cryo-EM datasets and a new version of relict / masking strategies to obtain high resolution maps and structures of dynactin's shoulder and pointed end complex. They propose sites at the pointed end that may be important for cargo adaptor binding, and find that these sites are conserved based on sequence analysis. This paper is a very good and thorough description of these two elements of the dynactin structure, which had been poorly resolved previously. Figures are very nicely prepared. We thank the authors for making this manuscript available to the community as a preprint on BioRxiv. As it was publicly available, we were able to discuss the manuscript in our lab's journal club, and comments below reflect contributions from several lab members.

Major points:

Our main critique of this manuscript is that we do not gain an understanding of how these new structures are functionally important. The structural work itself is very well described, and impressive, and the cryo EM data processing strategy is also interesting. However, their role in dynactin function or dynein motility is not elucidated. Some functional experiments may be necessary to follow up on the structural work, in order to understand how these structures help to further our understanding of dynactin function.

1) The authors propose four sites on the pointed end complex where cargo adaptors may bind. However, in the absence of followup mutations and functional assays it is unclear what role these sites play in adaptor binding. Similarly for the p150 arm, which is suggested to interact with sites 2, 3, and 4 of the pointed end, followup studies would be ideal.

2) An interesting finding of this study is the structural elucidation of p62's zinc binding motifs. The resolution of the densities allows the structural fold to be determined (i.e. one is a Gag-like knuckle and the other a zinc ribbon), but authors don't make any mention of the functional relevance of these domains. Since the structure of these motifs have now been solved, could the authors provide experimental data that elucidates their functional role, or some form of analysis regarding their function. What is known about these kinds of Zn-binding motifs in other biological contexts?

We do appreciate that the system is not trivial to work with, and that making and testing many mutations may not be feasible. However, some functional followup would substantially increase the impact of this study. In addition, could the authors use the results of published functional experiments, mutations, or disease mutations to contextualize their structure a bit more? This

would also provide more insight into how the structure enables more understanding into function.

Minor points:

- 1) Could the authors please provide details of how the data were merged? For example, when and how was the scaling factor applied?
- 2) The authors used an interesting approach to masking/signal subtraction, and we were curious how this compared with using the local resolution, multi-body refinement and/or 3D variability analysis to define the masks.
- 3) It would be helpful to add some labels onto the data processing workflow shown in Supplementary Fig. 1, and then correlate this with the description in the methods. Sometimes it was a bit hard to correlate the workflow with the methods section.
- 4) In Fig. 1A we feel it would be helpful to add in dynein, and perhaps dynein-dynactin-adaptor complex, to make the work more accessible, so the reader can easily orient how dynactin fits into the bigger picture
- 5) Fig. 2 - could the authors please show a comparison of the upper copy and lower copy, either side-by-side or using alignments/superpositions, as they determine would be best. We had a hard time visualizing the differences between the two copies.
- 6) Lines 221-223: "The central portion of p62, which we call the saddle, contains multiple cysteines that fold into three zinc-binding motifs, with density between the cysteines for metal ions." It would be helpful to refer to Figure 3A. Could the authors please show the density for the metal ions? In Table 1, it seems that Zn is built into this density, so it would be necessary to provide justification for building Zn into the model.
- 7) Lines 249-250: "It is shared by all three adaptors, with the loop appearing to adopt different conformations to bind each adaptor." It would be helpful if the authors could show a figure for this statement, as it is a bit hard to visualize.

Referee #3:

This study provides a refined structure of dynactin, a ≈ 1 MDa heteromeric complex. The authors apply novel masking strategies on pre-existing and new cryoEM datasets to obtain high resolution structures of the two "end" domains of dynactin - the shoulder and the pointed end. Without question, this is an impressive and valuable undertaking that is highly worthy of publication.

The study is in essence three smaller stories: one on shoulder structure, one on p62 structure, and one on the contacts between dynactin and a sampling of adaptors, plus an intramolecular contact between the projecting arm and the actin-like filament. Combined, these pieces provide a high resolution ($\approx 4\text{\AA}$) view of the entire dynactin complex, new information about the intermolecular contacts that allow dynactin to associate with cargoes, and new insight into dynactin autoinhibition. The work is a strong achievement given the importance of dynactin to cellular physiology and dynactin's unique and decidedly odd structure. The work is clean and largely convincing. However, given that this report effectively brings the dynactin structure story to a close,

I feel it is important that the authors address the following concerns.

General comment: It would greatly assist the reader in tracing the paths of p50, p24, p150 and p62 in the structures shown if simplified secondary structure diagrams were provided in the relevant figures. Detailed versions that include the primary sequences could be provided in the supplement (e.g., SF8).

I expect there is disorder at the N and C termini of p25, p27 and p62, and possibly p24. For all components, please specify the extent of the well-ordered regions (i.e., AA x to xxx).

Small points: Figure 1B is confusing. I am guessing the bottom of the rod corresponds to the pointed end but this should be specified.

Line 131: what is meant by "more meaningful priors for refinement"? Please be more clear/precise.

SF1 and Line 962: What is the artefactual density at the pointed end? How was it determined to be artefactual?

SF10: What does the dotted line signify?

Typos: Line 199: "flexible IN p50A"

Line 310: "CC1A/B"

Larger points:

Crosslinking data:

It is my understanding that the higher resolution achieved here allows the backbone to be generally traced, but does not allow identification of all side chains. There are also still some "gaps" in the structure, and many flexible "loops/linkers", suggesting some ambiguity remains in the assignments. To address this concern, the authors performed crosslinking mass-spec to identify contacts within and among protomers. This is a heroic and laudable effort, but mass spec analysis suffers from the usual concern that a negative result does not mean much. It seems possible that the amino composition of some components (i.e., few reactive amines for crosslinking, few lysines for tryptic digestion) would make some components less likely to show up in this analysis. The authors discuss the number of crosslinks identified (>500) and state that a small number are not consistent with the structure reported here. Diagrams showing crosslinks between and within some components are provided, but not all components are detailed. In the interest of transparency, rigor, and ruling out alternative interpretations, the full crosslinking data set should be made available.

The p62 structure looks very believable (and very odd - "unusual" line 220 is an understatement). By contrast, some details of the shoulder structure and the contacts between the adaptors and the filament give me pause. My detailed concerns follow.

Shoulder:

I found certain aspects of the description of the structure of the p50 N-terminal (hook/paddle) portions hard to follow. I realize that the shoulder is a novel and unusual structure and I am sorry that I don't have any suggestions, but I encourage the authors to make the description easier to understand. Given the flexibility of the connections between the hook/paddle and dimerization domain, and the proximity of both hooks to the dimerization domain, isn't it possible that the overall structure involves a domain swap where the upper arm leads into the lower hook/paddle and vice

versa? SF6 shows a discontinuity (long dotted line) connecting the paddle to the dimerization domain and other discontinuities are seen in Figure 2 (discussed further below).

Figure 2 is challenging. The structures in 2E and 2F trace the two different p50 protomers (A and B) but it's not immediately obvious how this relates to the structure shown in 2B and in the 2E inset. Maybe show the inset in the same orientation as in E and F? (The issue is how best to represent a 3D structure using 2D representation - this is no fault of the authors, but the reader needs more help here).

And what is the function of p24? I believe p24 has been reported to bind p150 directly in at least one study and p24 is required for p150 anchoring to the filament (shown genetically in yeast). The structure shows the shoulder component p24 to be contained completely within the shoulder arm, making no contribution to the paddle, hook or importantly, the dimerization domain. I am wondering how certain the authors are of this. The structure of the shoulder is very complex, with 8 alpha helices all feeding into the central dimerization domain. The upper and lower "copy" have slightly different structures, which I would think would impact resolution. It is stated that p50 "securely houses" p150, with p24 not participating. Was it possible to rigorously assign positions to the p24 N and C-termini? Do the crosslinking data shed any light on p24-p50-p150 interactions? All the helices contain non-helical "linker" segments which may have the potential to be "stretched" (see Figure 2E vs. F, dotted line in Figure 2D) so I don't understand why this might also be true for the linkers and kinks in p24 (Figure 2B and D). What AAs are in the loops in p50 and p24 (line 185; secondary structure representations would be helpful)? The authors admit that the side chain density is absent in the lower part of the hook structure which adds additional ambiguity (at least to my mind) to the overall structure of the arm.

There are gaps in the p150 density in the dimerization domain, and not all of the predicted secondary structural elements in this part of p150 are accounted for in the structure. [Again, a simplified secondary structure diagram of the p150 C-terminal portion would be very helpful.] Is it possible that two of the helices in the dimerization domain are contributed by p24? There are discontinuities in the structures shown in Figure 2C, D and F, which underscores the need to see how well the crosslinking data support the model. All of this is important because p150 is essential for dynactin function, so having an unimpeachable understanding of how it is anchored within the shoulder is highly worthwhile.

It is intriguing that p50 A and p50 B adopt distinct structures in the hook and paddle (Figure 2E/F). The authors state that this was verified by crosslinking, but I would like to see more details on this. Is this structure seen in both the upper and lower "copies", or just the lower "copy"? I'm a bit confused because the authors state the density is "worse" for the lower copy. [I encourage the authors to find another term for "copy", as this word is not particularly precise or scientific; also is "worse" density (line 895/6) standard terminology?]

Figure 2D shows a p50 linker contacting a p150 alpha helix (CC2?) Was this verified by crosslinking?

Is it possible to connect the paddle/hook to the p50 N-termini that bind Arp1? It seems that this is true for one of the "copies" but not the other. Again, the possibility of a domain swap seems possible, especially if there are gaps in density at important locations.

Pointed end proteins and adaptor contacts:

The p25 C-terminal alpha helix is not mentioned much. Previous available datasets (TDB, TDR,

TDH) suggests that the connection between this alpha helix and the core beta structure is flexible, as the precise position of the alpha helix depends upon which adaptor is bound. Given the problems with flexibility and averaging I am concerned this may impact interpretation. Please elaborate.

Are any of the interactions reported among Arp11, actin and the pointed end proteins p62, p25, p27 supported by crosslinking data? Given that this is not mentioned I would suppose not, but this should be clarified.

The authors identify interaction sites for adaptors and p150 on pointed end proteins and use alignments to implicate certain residues in p25 and p62 as being particularly important for these contacts. It would be very helpful if the authors would specify the complementary binding sites on the adaptors and p150 (perhaps AA 245-265, Line 301). Are the residues in BICD2 the same as those proposed in the modeling work of Zheng (2017)? I would think the registry of BICDR1 estimated on the position of W166 could be extended to identify contact residues with the dynactin pointed end. This is important information that will facilitate future studies work on how dynactin autoinhibition and adaptor binding are controlled.

The authors speculate that binding of BICDR1 and BICD2 to the end face of p25 is what induces the "kink" (line 379) seen in these adaptors. Can the authors discuss what adaptor residues constitute the kink- is it the so-called spindly motif that lies near the end of the coiled-coil portion of the adaptor?

Referee #1:

Cryo-EM reveals the complex architecture of dynein's shoulder and pointed end. Dynein is responsible for most of the minus-end directed transport along microtubules and is capable of transporting many different cargoes. To become processive, dynein must be activated by binding an activating adaptor protein and the multi-subunit protein complex dynactin. The structure of dynactin was determined to near atomic resolution several years ago, which allowed mapping of how dynein and adaptor proteins form a tripartite complex with dynactin. However, due to the flexible nature of the complex, several areas of dynactin were not resolved to high resolution, including the pointed end and shoulder regions, both of which are structurally and functionally important.

In this manuscript, Lau et al combine particles from prior cryo-EM datasets and reprocess the data to improve the resolution of the pointed end and shoulder domains. The authors provide a detailed description for their rationale behind choosing which masks are used for particle signal subtraction and local refinement. The higher resolution maps allow the authors to assign previously unknown densities to specific subunits of both the shoulder and pointed end. Subunit assignment was validated using crosslinking and mass spectrometry. Finally, atomic models were built for the shoulder and pointed end using their high-resolution maps. These models allowed the authors to show different adaptor proteins have distinct binding sites on the pointed end. Combining a modest increase in resolution for the docked conformation of p150 and their cross-linking data, the authors also propose that binding of p150 in that conformation would sterically compete with all three of the activating adaptors characterized to date (BICD2, BICDR1 and Hook3), adding another layer of regulation to an already baroque complex molecular motor.

This is a well-written, straight-forward description of a significant improvement in major components of dynactin, a central regulator of microtubule-based transport. This is an important advance of interest to a broad readership. The description of the data processing also makes the manuscript interesting to anyone in the cryo-EM community working on similarly complex systems. There are no major problems with the manuscript. All the comments below are minor suggestions I believe could help improve it.

Major issues

- None

Minor issues / Comments

- The authors describe sequence conservation between the adaptor proteins and interaction sites on the pointed end. Did they look to see if these sites show any evolution covariance? That might be more informative than looking at general sequence conservation of the individual proteins.

We have not yet been able to look if the adaptor sites and dynactin sites show evolutionary covariance. The only adaptor where we have an approximate registry estimate is BICDR1, which is conserved amongst vertebrates and insects. Unfortunately there were not enough paired pointed end:BICDR1 sequences to run a reliable evolution covariance analysis (Hopf et al. 2014). We will aim

to follow up on this suggestion in the future, when more sequences are available or when we can establish the registry of other adaptors.

- The authors performed CTF refinement in RELION 3.0. It would be interesting to see if the resolution improved after CTF refinement in RELION 3.1 using the higher order aberration and anisotropic magnification refinement options.

We performed CTF refinement with higher order aberration correction and magnification refinement in RELION 3.1 as requested, but came to the conclusion that neither strategy visually improved the final map.

Our original resolution of the shoulder map was 3.8 Å. Using magnification refinement in RELION 3.1 the output resolution was slightly worse (4.0 Å). Using CTF refinement with higher order aberration estimation, the output resolution was the same (3.8 Å).

- The new p150 docked data should be included in the Supplementary 1 processing figure.

We have now included this in a new Appendix Figure S13.

Why wasn't the new data included with the rest of the processing if particle count was a concern?

We tried including the new dynactin data, but it did not improve either the shoulder and pointed end. We think this is because dynactin on its own exhibits more flexibility in the shoulder and pointed end than the dynein-dynactin-adaptor datasets. We have mentioned this in the methods (line 679-681).

- How does particle subtraction compared with doing multibody refinement?

We tried multibody refinement with our optimized masks alongside particle subtraction. When compared to the equivalent stage, multibody refinement performed similarly for the shoulder (4.1 Å, compared to 4.1 Å in the manuscript), but worse for the pointed end (4.7 Å compared to 4.0 Å in the manuscript). This is probably because multi-body refinement does not allow recentering of the rigid bodies we define. This recentering during signal subtraction was important for our pointed end processing, helping us to improve the resolution of the pointed end. We have mentioned this in the methods (line 629-631).

- Figure 2A: It is slightly confusing that a strongly filtered map seems to be shown in this panel rather than the one that would have been used to build the model. Given that a major point of this paper is the improvement of the density, it would be less confusing to readers if a higher resolution map were shown here.

The representation is a surface representation of the model, rather than an electron density map. We feel that this rendering best shows how the large architectural features in the shoulder are organized. In the high resolution maps used for building the shoulder (shown in Figure 1D), some regions, particularly the upper paddle, are visible only as broken density. As suggested below for Fig.

3A/B, we have included a line in the figure legend describing how this figure was generated (line 1093).

- Figure 2B: I found the choice of colors confusing here as well. Blue, yellow and red are used in panel (A) to distinguish the Lower arm, Upper arm and dimerization domain. In panel (B), red and yellow are used again but this time to distinguish the proteins comprising the Lower arm, and the purple used for p150 is fairly close to the blue used in (A).

We have changed the colors in 2A to greens and different shades of blue to address this. Furthermore we have reworked Figure 2, splitting it into Figures 2-4 to better guide the reader through the structure, as part of a response to referees 2 and 3.

- Figure 2E: It is not possible to tell the handedness of the rotation from the symbol without introducing some breaks in the arrow/segment.

We have introduced breaks in the arrow/line segments as requested in all figures where the rotation symbol is included. Please note that this figure has moved to Figure 4D.

- Figure 3A/B: As with Figure 2, I found it confusing that the authors use a strongly filtered version of their maps rather than the one used for model building. Is the problem that the level of detail in the higher-resolution map distracts from the ribbon diagrams? Would a more transparent setting help with that? If the authors feel this is the best way of presenting things, they may want to add a line to the figure legend (here and in Figure 2) explaining that the surface representation has been filtered for the purposes of the figure.

The level of detail in the higher-resolution maps does indeed distract from the ribbon diagrams here (now renamed Figure 5). We have included a line in the figure legend describing how this figure was generated (line 1129).

- Figure 4E: The color choices in this panel are also potentially confusing as green and purple are used for amino acids in panels (C) and (D).

We have changed the amino acid colors in 4C and 4D to remove green and purple. Please note these panels have moved to Figure EV3.

- Figure 5C: It would be nice to include in this panel the same dashed boxes shown in Figure 4B for binding sites 2-4 (this would prevent the reader from having to go back and forth between them).

We have added the dashed boxes denoting sites 2-4 to figure 5C (now Figure 7C) as requested.

Very minor syntax point

- Line 27: "Dynactin is a 1.1 MDa complex that activates the molecular motor, dynein, for ultra-processive..." and Line 42: Dynactin is a large, multi-subunit co-activator for the molecular motor, cytoplasmic dynein 1." It seems to me that the commas in "...dynein,..." and "..., cytoplasmic dynein

1" should not be there. "Dynein" and "cytoplasmic dynein 1" are not clauses that can be removed from those sentences without changing their meaning, as dynein is not the only molecular motor. Only "dynein" or "the molecular motor dynein" would uniquely identify it.

We agree with the referee, and have removed these commas.

Referee #2:

Summary:

In this work, the authors take advantage of three cryo-EM datasets and a new version of relion / masking strategies to obtain high resolution maps and structures of dynactin's shoulder and pointed end complex. They propose sites at the pointed end that may be important for cargo adaptor binding, and find that these sites are conserved based on sequence analysis. This paper is a very good and thorough description of these two elements of the dynactin structure, which had been poorly resolved previously. Figures are very nicely prepared. We thank the authors for making this manuscript available to the community as a preprint on BioRxiv. As it was publicly available, we were able to discuss the manuscript in our lab's journal club, and comments below reflect contributions from several lab members.

Major points:

Our main critique of this manuscript is that we do not gain an understanding of how these new structures are functionally important. The structural work itself is very well described, and impressive, and the cryo EM data processing strategy is also interesting. However, their role in dynactin function or dynein motility is not elucidated. Some functional experiments may be necessary to follow up on the structural work, in order to understand how these structures help to further our understanding of dynactin function.

1) The authors propose four sites on the pointed end complex where cargo adaptors may bind. However, in the absence of followup mutations and functional assays it is unclear what role these sites play in adaptor binding. Similarly for the p150 arm, which is suggested to interact with sites 2, 3, and 4 of the pointed end, followup studies would be ideal.

Making site-specific mutations in a full dynactin complex is a considerable undertaking, as a recombinant system for mutating and expressing the full dynactin complex is not yet easily available. We therefore cloned and expressed an isolated pointed end construct comprising ZZ-tagged Arp11, p62, p25 and p27. This pointed end construct has previously been shown to interact with Strep-tagged cargo adaptors Spindly and BICD2 (Gama et al. 2017).

We created pointed end mutants of each binding site. For Site 1, we replaced the positively-charged region of the loop to a Gly-Ser linker of identical length. For Sites 2, 3 and 4, we mutated all the interacting residues in each site to alanine.

To get the pulldown assay to work, we cloned BICD2(1-400) and BICDR1 to remove the ZZ-tag and replace it with a Strep-tag. We also expressed and purified Strep-Hook3(1-522) (Urnavicius et al. 2018). We successfully purified Strep-Hook3 and Strep-BICD2, but Strep-BICDR1 showed considerable degradation in both insect cells and bacteria. We hence proceeded with Strep-Hook3 and Strep-BICD2.

Our assay shows that Hook3 and BICD2 bind weakly to the pointed end when sites 1 or 4 are mutated. In contrast mutation of sites 2 and 3 had less or no effect. These results show the importance of the flexible positively-charged site 1 loop for adaptor binding. We did not expect Hook3 to bind to site 4 as its density was weak in this region of our EM maps. However, we noticed on reexamination that there is extra density, consistent with part of Hook3 interacting with the Site 4 - perhaps a helix connected by a flexible linker to one of the main coiled coils (Figure R1 below). Overall, the structure and mutagenesis reveal that sites 1 and 4 are the critical points of adaptor recognition on the pointed end complex. Future work will attempt to improve the resolution of the site 4 with cargo adaptor bound in order to understand the basis of adaptor binding here.

We have added these pulldowns in the results (lines 307-315, Figure 6D,E, Appendix Figure S12), and have reworked the discussion to include the insights gained from this experiment (from line 433). We have described the cloning, purification and pulldown protocols in the methods (from lines 494 and 844 respectively)

To understand how p150 competed with adaptor binding to pointed end complexes. To do this we attempted to express p150 constructs consisting of CC1A/B (residues 214-547). However we were

not able to not get the expression and pulldowns to work satisfactorily. We aim to pursue this in future work.

2) An interesting finding of this study is the structural elucidation of p62's zinc binding motifs. The resolution of the densities allows the structural fold to be determined (i.e. one is a Gag-like knuckle and the other a zinc ribbon), but authors don't make any mention of the functional relevance of these domains. Since the structure of these motifs have now been solved, could the authors provide experimental data that elucidates their functional role, or some form of analysis regarding their function. What is known about these kinds of Zn-binding motifs in other biological contexts?

Our structure suggests that the zinc-binding motifs play a structural role in p62. Indeed similar sites in other proteins are often structural in nature (Krishna et al. 2003; Lee et al. 2008), and these proteins fold improperly if the zincs are removed (Kluska et al. 2018). We have briefly mentioned the structural role of our zinc-binding motifs in the manuscript (line 398-400). Please note that when re-evaluating our structure, we realized that the motif we previously identified as a Gag-like knuckle was better described as a second zinc ribbon (line 397)(Krishna et al. 2003).

We do appreciate that the system is not trivial to work with, and that making and testing many mutations may not be feasible. However, some functional followup would substantially increase the impact of this study. In addition, could the authors use the results of published functional experiments, mutations, or disease mutations to contextualize their structure a bit more? This would also provide more insight into how the structure enables more understanding into function.

The pointed end pulldown assay we performed has been informative, and hence we include these experiments in the results. These assays have highlighted the importance of site 4 for cargo adaptor binding. In our manuscript, we mentioned that site 4 contains Serine 31, which has been identified as a phosphorylation site. With our new pulldown assay results, we can now speculate that this phosphorylation site may represent a key control mechanism for adaptor recruitment. We mention this in the discussion (line 472-474).

There is only one mutation in the Cancer Genome Atlas (Hutter et al. 2018) that is found in any of our pointed end sites - Site 4 (residue V35, mutating to isoleucine). However as this mutation has not yet been causally linked to cancer we have not discussed it in the current manuscript.

Minor points:

1) Could the authors please provide details of how the data were merged? For example, when and how was the scaling factor applied?

We used the scaling method described previously (Wilkinson et al. 2019) at two points in our processing. The first was to merge the TDH datasets (LMB K2 dataset, Leeds dataset and the dataset from Urnavicius et al. 2018). The second was to merge the TDH dataset with the TDR/dynactin datasets. We now include a brief description of the scaling process in the methods (line 572-578).

2) The authors used an interesting approach to masking/signal subtraction, and we were curious

how this compared with using the local resolution, multi-body refinement and/or 3D variability analysis to define the masks.

We tried multibody refinement using our optimized masks. However it performed worse for the pointed end, compared with our approach (see response to referee 1). The output did show us how the rigid bodies that we defined (the shoulder region and pointed end) moved relative to one another, but the maps and motion were too coarse to see the level of detail needed to optimize the masks.

We calculated local resolution of the map we used to optimize our shoulder mask, to assess if this could be used to help mask optimization. This identified similar regions to add/remove to the mask, as we did in our optimization. We have mentioned this in the methods (line 608-609). It would be interesting in future work to see whether one can create a script to iteratively optimize a mask based on local resolution, though this is beyond the scope of this study.

3) It would be helpful to add some labels onto the data processing workflow shown in Supplementary Fig. 1, and then correlate this with the description in the methods. Sometimes it was a bit hard to correlate the workflow with the methods section.

We thank the referee for this excellent suggestion. We have added labels A-N in Supplementary Figure 1 (renamed Figure EV1) as reference points for the methods.

4) In Fig. 1A we feel it would be helpful to add in dynein, and perhaps dynein-dynactin-adaptor complex, to make the work more accessible, so the reader can easily orient how dynactin fits into the bigger picture

We have added a new Figure 1A to put dynactin into context with dynein, an adaptor and a microtubule.

5) Fig. 2 - could the authors please show a comparison of the upper copy and lower copy, either side-by-side or using alignments/superpositions, as they determine would be best. We had a hard time visualizing the differences between the two copies.

We have included side-by-side comparisons of the upper and lower copy of the shoulder in Figure 2 (panels B and C). More broadly, we have split figure 2 into figures 2-4 to better guide the reader through our shoulder structure.

6) Lines 221-223: "The central portion of p62, which we call the saddle, contains multiple cysteines that fold into three zinc-binding motifs, with density between the cysteines for metal ions." It would be helpful to refer to Figure 3A.

We have included the suggested figure reference, pointing to Figure 3A (now Figure 5A).

Could the authors please show the density for the metal ions? In Table 1, it seems that Zn is built into this density, so it would be necessary to provide justification for building Zn into the model.

We have shown the density for the metal ions in Appendix Figure S9C. The motifs in these sites are known to predominantly bind zinc, but in rare cases coordinate iron. We have updated the manuscript to explain why we assign zincs (line 242-244).

7) Lines 249-250: "It is shared by all three adaptors, with the loop appearing to adopt different conformations to bind each adaptor." It would be helpful if the authors could show a figure for this statement, as it is a bit hard to visualize.

We have added Figure EV3A. These show orthogonal views compared to Figure 6A. Because the loop is hidden behind the adaptors, we show the adaptors in white transparency. The ordered density of the loop is different when interacting with different adaptors, suggesting different conformations. We have highlighted the ordered part of the loop with a black outline in each case.

Referee #3:

This study provides a refined structure of dynactin, a ≈ 1 MDa heteromeric complex. The authors apply novel masking strategies on pre-existing and new cryoEM datasets to obtain high resolution structures of the two "end" domains of dynactin - the shoulder and the pointed end. Without question, this is an impressive and valuable undertaking that is highly worthy of publication.

The study is in essence three smaller stories: one on shoulder structure, one on p62 structure, and one on the contacts between dynactin and a sampling of adaptors, plus an intramolecular contact between the projecting arm and the actin-like filament. Combined, these pieces provide a high resolution ($\approx 4\text{\AA}$) view of the entire dynactin complex, new information about the intermolecular contacts that allow dynactin to associate with cargoes, and new insight into dynactin autoinhibition. The work is a strong achievement given the importance of dynactin to cellular physiology and dynactin's unique and decidedly odd structure. The work is clean and largely convincing. However, given that this report effectively brings the dynactin structure story to a close, I feel it is important that the authors address the following concerns.

All referees referred to Figure 2 as being difficult to follow. It was helpful to have this pointed out. To address their comments and assist the reader in understanding the shoulder structure, we have split the current Figure 2 into Figures 2-4 and have made a number of improvements. Please note that we have also replaced the word "copy" with "subdomain", in response to the referee comment below.

The new Figure 2 shows an overview of how the upper and lower subdomains in the shoulder sit on the filament (in response to referee 2). The new Figure 3 shows the paths of different chains within the lower subdomain. Figure 4 shows the alternative paths of p50-A and -B using rainbow coloring with secondary structure diagrams. The paths of p150 and p24 are also shown in the same representation in Figure EV2.

General comment: It would greatly assist the reader in tracing the paths of p50, p24, p150 and p62 in the structures shown if simplified secondary structure diagrams were provided in the relevant figures. Detailed versions that include the primary sequences could be provided in the supplement (e.g., SF8).

We have included simplified secondary structure diagrams in the new Figure 3, 4 and Figure EV2 for the shoulder subunits, and have included a secondary structure figure for p62 in Figure 5. We have included primary sequence with secondary structure assignment in the supplementary figures (Appendix Figures S5, S6 and S10).

I expect there is disorder at the N and C termini of p25, p27 and p62, and possibly p24. For all components, please specify the extent of the well-ordered regions (i.e., AA x to xxx).

We have specified the well-ordered regions in the methods (line 697-723). We summarize which parts are disordered, including the residues at the N- and C-termini, in Appendix Figures S5, S6 and S10.

Small points: Figure 1B is confusing. I am guessing the bottom of the rod corresponds to the pointed end but this should be specified.

Figure 1B (now Figure 1C) shows density corresponding to an alpha helix in dynactin's shoulder, to illustrate the improvement in resolution. We have tried to clarify this by including a cartoon in the figure to show the position of the helix within the shoulder.

Line 131: what is meant by "more meaningful priors for refinement"? Please be more clear/precise.

We now add an explanation in the text (line 129). When you recenter a particle stack, you reduce the errors in the assignment of rotations and offsets of each particle, relative to the consensus model.

SF1 and Line 962: What is the artefactual density at the pointed end? How was it determined to be artefactual?

The artefactual density was caused by the original mask, used to subtract the dyneins and the adaptors, cutting into the β -sandwich fold of p62. The striated appearance of this density is typical of that seen when a soft-edged mask results in the partial subtraction of a structure. We therefore re-performed the subtraction of the dyneins and adaptors with a better designed mask, producing better density for the β -sandwich fold. We have clarified this in Figure EV1, the legend of Figure EV1 and in the methods (lines 635-640).

SF10: What does the dotted line signify?

The dotted line Supplementary Figure 10 (now Appendix Figure S18) was originally meant to signify the approximate location of the crosslinks that are consistent with CC1A folding back to contact

CC1B (red in the lower panel of the figure). To make this clearer, we have removed the dotted line and represented these crosslinks directly on the cartoon in the figure.

Typos: Line 199: "flexible IN p50A"

Line 310: "CC1A/B"

We have corrected these typos.

Larger points:

Crosslinking data:

It is my understanding that the higher resolution achieved here allows the backbone to be generally traced, but does not allow identification of all side chains. There are also still some "gaps" in the structure, and many flexible "loops/linkers", suggesting some ambiguity remains in the assignments. To address this concern, the authors performed crosslinking mass-spec to identify contacts within and among protomers. This is a heroic and laudable effort, but mass spec analysis suffers from the usual concern that a negative result does not mean much. It seems possible that the amino composition of some components (i.e., few reactive amines for crosslinking, few lysines for tryptic digestion) would make some components less likely to show up in this analysis. The authors discuss the number of crosslinks identified (>500) and state that a small number are not consistent with the structure reported here. Diagrams showing crosslinks between and within some components are provided, but not all components are detailed. In the interest of transparency, rigor, and ruling out alternative interpretations, the full crosslinking data set should be made available.

The crosslinking dataset was uploaded to the PRIDE dataset (PXD020084). We have now added the reference to the main text (line 140). It is currently available for the referees (referee username reviewer32101@ebi.ac.uk and password 65MHxtSc), and will be released along with the manuscript upon publication.

The p62 structure looks very believable (and very odd - "unusual" line 220 is an understatement). By contrast, some details of the shoulder structure and the contacts between the adaptors and the filament give me pause. My detailed concerns follow.

Shoulder:

I found certain aspects of the description of the structure of the p50 N-terminal (hook/paddle) portions hard to follow. I realize that the shoulder is a novel and unusual structure and I am sorry that I don't have any suggestions, but I encourage the authors to make the description easier to understand. Given the flexibility of the connections between the hook/paddle and dimerization domain, and the proximity of both hooks to the dimerization domain, isn't it possible that the overall structure involves a domain swap where the upper arm leads into the lower hook/paddle and vice versa?

We thank the referee for drawing our attention to this. We have substantially reworked the whole structure description of the shoulder to make it easier to understand (lines 150-226). Specifically for p50, we now describe paths of the two copies (p50-A and p50-B) running from their similar C termini

to the divergent N termini (lines 178-215). We also number the helices in this subunit to allow us to refer to structural differences between p50-A and p50-B.

The quality of our maps allows us to be confident of the assignment of the shoulder subunits in our model. Briefly, the maps have good connectivity at lower threshold allowing us to trace the complete paths of all subunits. In addition, we have coverage for all the main regions of the structure at sidechain resolution, which allows us to assign the subunits. We now explain this at the start of the structure section (lines 158-167). We also include a new supplementary figure (Appendix Figure S7), which shows the parts of the structure for which we have sidechain resolution and the parts that allow us trace the connectivity. It also shows the eight loops for which there is little or no density. However in all of these cases the equivalent loop is visible in another copy of the same subunit.

SF6 shows a discontinuity (long dotted line) connecting the paddle to the dimerization domain and other discontinuities are seen in Figure 2 (discussed further below).

We use dotted lines in two contexts. First they can refer to regions of the structure where we can see connections at low threshold. This is the case of the loop shown in SF6 (now SF 10). Second, they can refer to loops which are not visible due to their flexibility, but whose connectivity we can be confident of.

We have now included secondary structure diagrams mapped onto primary sequence (Appendix Figures S5 and S6), where we annotate parts of the structure for which we have no density in any copy (light gray dashes).

Figure 2 is challenging. The structures in 2E and 2F trace the two different p50 protomers (A and B) but it's not immediately obvious how this relates to the structure shown in 2B and in the 2E inset. Maybe show the inset in the same orientation as in E and F? (The issue is how best to represent a 3D structure using 2D representation - this is no fault of the authors, but the reader needs more help here).

As described above, we have reworked Figure 2 into three new figures (Figures 2-4) to help the reader. The new Figure 4 now includes the whole of p50-A and p50-B in rainbow coloring to help orient the reader. The previous figures 2E and F are now shown in Figure 4D and E.

We have also included a new Figure EV2 which shows the paths of p24 and p150 (C terminal domain) using rainbow coloring.

And what is the function of p24? I believe p24 has been reported to bind p150 directly in at least one study and p24 is required for p150 anchoring to the filament (shown genetically in yeast). The structure shows the shoulder component p24 to be contained completely within the shoulder arm, making no contribution to the paddle, hook or importantly, the dimerization domain. I am wondering how certain the authors are of this. The structure of the shoulder is very complex, with 8 alpha helices all feeding into the central dimerization domain. The upper and lower "copy" have slightly different structures, which I would think would impact resolution.

As detailed elsewhere in this response, we are confident in our placement of p24. Although p24 and p150 makes some direct interactions, our structure suggests that p24's main function is ensuring the correct folding of the shoulder. Taken together with the studies cited by the referee, this suggests that the integrity of the whole shoulder is essential for the correct anchoring of p150. We now include the studies cited and discuss this in the manuscript (lines 375-380).

It is stated that p50 "securely houses" p150, with p24 not participating.

Apologies for the misunderstanding - in our abstract, the element that "securely houses p150" is unclear. We have now reworded this section (line 35), to clarify that it is the unique architecture of the shoulder that securely houses the p150, rather than the p50 specifically.

Was it possible to rigorously assign positions to the p24 N and C-termini?

We can trace residues 4-173 of p24 in both the lower and upper tetramers, including the linkers. We include this information in Appendix Figures S5 and S7. p24 is 187 amino acids long. We do not see density for the N terminal and C terminal residues, even at low threshold, suggesting they are flexible and disordered. Although we cannot precisely place these terminal residues, there is not enough polypeptide for p24 to contribute to other regions of the shoulder.

Do the crosslinking data shed any light on p24-p50-p150 interactions?

Previously we noted that the number of crosslinks that were overlength was below our false discovery rate, which means that the crosslinking data is generally compatible with the model we built.

We now include in the manuscript three specific examples of how crosslinking data shed light on the p24-p50-p150 interactions. Crosslinks from p150 as it enters the shoulder validate the assigned registries of p24 and p50 in the helical arms (Appendix Figure S8A, lines 169-171). Crosslinks between p24 and p50 in the same region further support their arrangement (dashed lines in Appendix Figure S8A, lines 188-189). Crosslinks between p50-A and -B, and p24 provide support for the alternative conformations of the N terminal portions of the two p50s in each subdomain (Appendix Figure S9D and E, lines 210-215).

All the helices contain non-helical "linker" segments which may have the potential to be "stretched" (see Figure 2E vs. F, dotted line in Figure 2D) so I don't understand why this might also be true for the linkers and kinks in p24 (Figure 2B and D). What AAs are in the loops in p50 and p24 (line 185; secondary structure representations would be helpful)? The authors admit that the side chain density is absent in the lower part of the hook structure which adds additional ambiguity (at least to my mind) to the overall structure of the arm.

We now include secondary structure assignments mapped onto primary sequence in Appendix Figure S5. We also include the amino acid numbers of the helical break in Figure 2D (now 3C) in the main text (line 191-193), and map them on the simplified secondary structure in 2E (# symbol).

In p24, there are 4 linkers. We can see good density for the linker in the helical break (Appendix Figure S8B). The other three linkers are in the middle of the arm. There is enough density here to allow us to trace the connections. We assigned their backbone based on sequence of the surrounding helices, where we can see and assign sidechains.

There are gaps in the p150 density in the dimerization domain, and not all of the predicted secondary structural elements in this part of p150 are accounted for in the structure. [Again, a simplified secondary structure diagram of the p150 C-terminal portion would be very helpful.] Is it possible that two of the helices in the dimerization domain are contributed by p24? There are discontinuities in the structures shown in Figure 2C, D and F, which underscores the need to see how well the crosslinking data support the model. All of this is important because p150 is essential for dynactin function, so having an unimpeachable understanding of how it is anchored within the shoulder is highly worthwhile.

In the lower subdomain, we can see continuous density for the entirety of the p150 C terminal domain (residues 1096-1286). We have clarified this in the methods (line 716), and have illustrated it in Appendix Figure S7, which shows the parts of the structure that can be traced. We also include this information in our primary sequence-secondary structure diagram of p150 in Appendix Figure S6.

In the dimerization domain each of the four helices has sidechain density. Together with the connectivity we can trace, this allows to unambiguously assign them to p150 and p50. We have clarified this in the methods (lines 697-709), and shown it in Appendix Figure S7.

It is intriguing that p50 A and p50 B adopt distinct structures in the hook and paddle (Figure 2E/F). The authors state that this was verified by crosslinking, but I would like to see more details on this. Is this structure seen in both the upper and lower "copies", or just the lower "copy"? I'm a bit confused because the authors state the density is "worse" for the lower copy. [I encourage the authors to find another term for "copy", as this word is not particularly precise or scientific; also is "worse" density (line 895/6) standard terminology?]

As suggested, we have now changed the term "copy" to "subdomain". We have also replaced "worse" with "lower resolution".

As described above, we now include new Appendix Figures S8D and E, which illustrate how crosslinking data supports our model for alternative positions of p50-A and -B. We refer to this in the text (line 210-215).

The same arrangement of p50-A and -B is indeed observed in both subdomains. Although the density is at lower resolution in parts of the upper subdomain, we can clearly identify the same secondary structure elements in both subdomains. This shows that they are equivalent to one another. We now clearly state in the manuscript (lines 164-166).

Figure 2D shows a p50 linker contacting a p150 alpha helix (CC2?) Was this verified by crosslinking?

There are no crosslinks between this p50 linker and the p150 alpha helix. This is probably because this interaction is in the interior of the protein. However, the p50 linker fits into clear density in our EM maps, with the preceding and following helices assigned using sidechain density, which allows us to be confident of the interaction. We now include this fit of the p50-A linker into density in our new Appendix Figure S8B.

Is it possible to connect the paddle/hook to the p50 N-termini that bind Arp1? It seems that this is true for one of the "copies" but not the other. Again, the possibility of a domain swap seems possible, especially if there are gaps in density at important locations.

For the lower subdomain, we can clearly connect both p50s to the N termini contacting the Arp1 filament. For the upper subdomain, the density is ambiguous and so we only tentatively assign the connections. We have now included a description of which p50 binds to which filament-bound N terminus in the results section (lines 217-226), and in the methods (lines 725-730).

Pointed end proteins and adaptor contacts:

The p25 C-terminal alpha helix is not mentioned much. Previous available datasets (TDB, TDR, TDH) suggests that the connection between this alpha helix and the core beta structure is flexible, as the precise position of the alpha helix depends upon which adaptor is bound. Given the problems with flexibility and averaging I am concerned this may impact interpretation. Please elaborate.

We think that the p25 C-terminal alpha helix is not flexible relative to the core beta structure. We provide a figure here to support this (Figure R2). In our TDR (left) and TDH (right) structures, the density shows that the alpha helix is in the same position relative to the beta sheet structure, albeit at lower resolution. In our combined structure, we see density for the helical pitch and sidechains in the C terminal helix. This is because the TDR, TDH and dynactin datasets align well in this region. We have now mentioned that this helix is rigidly attached to the beta-helical fold (line 253-254).

Figure R2: A comparison between Tail:Dynactin:BICDR1 (TDR) and Tail:Dynactin:Hook3 (TDH, right), focused on the p25.

Are any of the interactions reported among Arp11, actin and the pointed end proteins p62, p25, p27

supported by crosslinking data? Given that this is not mentioned I would suppose not, but this should be clarified.

Our model of the pointed end complex is supported by 39 crosslinks, 12 of which occur between subunits. Two are between p62 and actin, four between p25 and p62, and six between arp11 and p62. We have now specifically mentioned the number of crosslinks supporting our structure in the manuscript (lines 233), and included an enlarged overview of the pointed end crosslinks in Appendix Figure S4C. Furthermore, we have included two specific examples in the manuscript that illustrate our subunit assignments. The first example is between p25 C terminal helix and p62 beta sandwich (line 238-239, Appendix Figure S9B). The second example is between the p62 long helix-loop and actin (line 249-250, Appendix Figure S11A).

The authors identify interaction sites for adaptors and p150 on pointed end proteins and use alignments to implicate certain residues in p25 and p62 as being particularly important for these contacts. It would be very helpful if the authors would specify the complementary binding sites on the adaptors and p150 (perhaps AA 245-265, Line 301). Are the residues in BICD2 the same as those proposed in the modeling work of Zheng (2017)? I would think the registry of BICDR1 estimated on the position of W166 could be extended to identify contact residues with the dynactin pointed end. This is important information that will facilitate future studies work on how dynactin autoinhibition and adaptor binding are controlled.

The authors speculate that binding of BICDR1 and BICD2 to the end face of p25 is what induces the "kink" (line 379) seen in these adaptors. Can the authors discuss what adaptor residues constitute the kink- is it the so-called spindly motif that lies near the end of the coiled-coil portion of the adaptor?

To address this comment, we have now expanded our discussion to include a section on how cargo adaptors bind dynein-dynactin (starting on line 441). As requested, we use the registry of BICDR1, estimated from the position of W166, to suggest the function of the different interaction motifs found in this adaptor. We propose that a previously-identified CC2 box (Sacristan et al. 2018), found close to the well-characterized CC1 box, interacts with a dynein heavy chain. We identify the stretch of negatively-charged residues that probably interacts with Site 1. Finally we find that the Spindly box is near the kink of BICDR1, and is indeed likely to interact with site 4.

Although we discuss the possibility that the BICD2 CC1 box and CC2 box could align with those in BICDR1, we cannot be sure of this. Therefore we do not feel that we can talk about residue-level interactions in BICD2 at the current time.

For p150 binding to the pointed end, the resolution of CC1A and CC1B is also low. We have made an approximation at the registry, which we use to illustrate the position of the crosslinking data (Figure 7B). Although we don't feel we can predict exact residues on p150 that interact, we have now specified on Appendix Figure S6 those residues to which we detected crosslinks to p62 and p25.

References

- Gama, J. B. et al. 2017. Molecular Mechanism of Dynein Recruitment to Kinetochores by the Rod-Zw10-Zwilch Complex and Spindly. *The Journal of Cell Biology* 216(4): 943–60.
- Hopf, T. A. et al. 2014. Sequence Co-Evolution Gives 3D Contacts and Structures of Protein Complexes. *eLife* 3.
- Hutter, C., and Zenklusen, J. C. 2018. The Cancer Genome Atlas: Creating Lasting Value beyond Its Data. *Cell* 173(2): 283–85.
- Kluska, K., Adamczyk, J., and Krężel, A. 2018. Metal Binding Properties, Stability and Reactivity of Zinc Fingers. *Coordination Chemistry Reviews* 367: 18–64.
- Krishna, S. S., Majumdar, I., and Grishin, N. V. 2003. Structural Classification of Zinc Fingers: SURVEY AND SUMMARY. *Nucleic Acids Research* 31(2): 532–50.
- Lee, Y., and Lim, C. 2008. Physical Basis of Structural and Catalytic Zn-Binding Sites in Proteins. *Journal of Molecular Biology* 379(3): 545–53.
- Sacristan, C. et al. 2018. Dynamic Kinetochore Size Regulation Promotes Microtubule Capture and Chromosome Biorientation in Mitosis. *Nature Cell Biology* 20(7): 800–810.
- Urnavicius, L. et al. 2018. Cryo-EM Shows How Dynactin Recruits Two Dyneins for Faster Movement. *Nature* 554(7691): 202–6.
- Wilkinson, M. E., Kumar, A., and Casañal, A. 2019. Methods for Merging Data Sets in Electron Cryo-Microscopy. *Acta Crystallographica Section D Structural Biology* 75(9): 782–91.

Thank you for submitting a revised version of your manuscript. I apologise for the delay in the processing of your manuscript due to the high number of submissions we are handling at the moment. Your revised study has now been seen by two of the original referees, who find that their main concerns have been addressed and recommend publication of the manuscript pending minor revision. Therefore, I would like to invite you to address the remaining minor referee comments and the following editorial issues.

Please let me know if you have any further questions regarding any of these points. You can use the link below to upload the revised files.

Referee #2:

We thank the authors for addressing all reviewer comments thoroughly and carefully. The pulldown experiments add functional insights to this work, and help to put the structure in context. One control that we feel should be included in relation to the pulldown experiments is some means of showing that the mutants themselves are stable and well-behaved - size exclusion chromatograms, some other part of the purification, or any data the authors feel could speak to this. Once these data are included, we feel that all comments will have been addressed. We congratulate the authors on a very nice paper.

Referee #3:

This revised manuscript does a great job of addressing all my previous concerns. This is an important and outstanding piece of work. The addition of details of the crosslinks clarifies and strengthens the story. The figures have been revamped and extended in a way that helps the reader walk through this unusual and interesting structure. I ask that the authors please to address the following minor concerns in the interest of clarity and completeness.

Minor issues:

P62 exists in two isoforms that differ by an exon encoding 7 AA that maps within to the disordered loop (detailed in Figure EV4). The alignment (Fig. EV4), showing the shorter human isoform, reveals that some species lack these residues entirely. This raises the possibility that this part of the protein might provide for additional functions/regulatory mechanisms. Given the importance of p62 site 1 for adaptor binding it would be helpful if the authors would consider and comment on this possible role of this alternative exon.

The addition of pulldowns with mutagenized recombinant pointed end complex is very nice (Figure 6D/E and S12). I myself trust the rigor of this work, but the authors might want to augment Figure S12 to include lane(s) of the purified complex(es) so the reader can see for themselves that the sample is reasonably pure and that the mutations don't alter complex composition or stability (I assume they don't; this would be worth specifying). I realize the authors were working with very low amounts of protein so this may not be readily doable.

Very minor suggestions:

Lines 213, 215: I believe the authors mean Figure S8E, not S10E

Line 234: Figure S4C instead of Appendix Fig. 5C.

Line 460: "flexibility" implies structural flexibility to me so this may not be the best word. choice. I think the authors mean to communicate that the sequence is variable.

Please annotate the secondary structure diagrams of p50 in Figures 3 and 4 with at least one additional residue number to help the reader identify the locations of S1 and H4. I realize the details are in Figure S5 but there are a lot of figures for the reader to digest - at the very least please refer the reader to S5 for additional detail.

In Figure 5, is it possible to label the Zn²⁺ (1, 2 and 3?) to help the reader match each the corresponding sequence element (specified in Figure S10)). This will facilitate orientation to the overall structural features of p62 which, to my knowledge, is an unusual structure.

Figure S4B: Please explain in the legend what the purple color indicates.

Referee #2:

We thank the authors for addressing all reviewer comments thoroughly and carefully. The pulldown experiments add functional insights to this work, and help to put the structure in context. One control that we feel should be included in relation to the pulldown experiments is some means of showing that the mutants themselves are stable and well-behaved - size exclusion chromatograms, some other part of the purification, or any data the authors feel could speak to this. Once these data are included, we feel that all comments will have been addressed. We congratulate the authors on a very nice paper.

The wildtype pointed end construct and the four mutant constructs all behaved well, eluting as monodisperse peaks from size exclusion chromatography containing all subunits. We have included SDS-PAGE gels and gel filtration traces for the five pointed end constructs used in Appendix Figure S12 (A and B) to address this comment and a similar one from referee 3. We have also addressed this in the text (line 313).

Referee #3:

This revised manuscript does a great job of addressing all my previous concerns. This is an important and outstanding piece of work. The addition of details of the crosslinks clarifies and strengthens the story. The figures have been revamped and extended in a way that helps the reader walk through this unusual and interesting structure. I ask that the authors please to address the following minor concerns in the interest of clarity and completeness.

Minor issues:

P62 exists in two isoforms that differ by an exon encoding 7 AA that maps within to the disordered loop (detailed in Figure EV4). The alignment (Fig. EV4), showing the shorter human isoform, reveals that some species lack these residues entirely. This raises the possibility that this part of the protein might provide for additional functions/regulatory mechanisms. Given the importance of p62 site 1 for adaptor binding It would be helpful if the authors would consider and comment on this possible role of this alternative exon.

The p62 alternatively-spliced exon encodes seven amino acids ("QHTIHVV"), and is conserved between vertebrates (Hammesfahr et al. 2012). We now show both isoforms in the human sequence in Figure EV4, and highlighted this short section in a dashed black box. Our recombinant human p62 construct does not include this exon (now specified in the methods, line 505), showing it is not required for adaptor binding. This short sequence does not change the charge of site 1 but may indeed have additional functions such as regulating vertebrate-specific adaptors. We now mention this in the discussion (lines 464-467).

The addition of pulldowns with mutagenized recombinant pointed end complex is very nice (Figure 6D/E and S12). I myself trust the rigor of this work, but the authors might want to augment Figure S12 to include lane(s) of the purified complex(es) so the reader can see for themselves that the sample is reasonably pure and that the mutations don't alter complex composition or stability (I assume

they don't; this would be worth specifying). I realize the authors were working with very low amounts of protein so this may not be readily doable.

The purified pointed end complexes each contained all subunits and behave well over size exclusion chromatography. We have included SDS-PAGE gels and gel filtration in Appendix Figure S12 (A and B) to address this comment and a similar one from referee 2. We have also specified this in the text (line 313).

Very minor suggestions:

Lines 213, 215: I believe the authors mean Figure S8E, not S10E

Line 234: Figure S4C instead of Appendix Fig. 5C.

We have corrected these figure references.

Line 460: "flexibility" implies structural flexibility to me so this may not be the best word. choice. I think the authors mean to communicate that the sequence is variable.

Here we are referring to the structural flexibility (shown in Figure EV3), which points to the precise location of the binding site on each adaptor will vary in position. We realized this was not previously clear and have modified that sentence to read "The structural flexibility of the site 1 loop (Fig EV3A) indicates that the precise location of its interaction site on each adaptor will differ" (now line 463-464).

Please annotate the secondary structure diagrams of p50 in Figures 3 and 4 with at least one additional residue number to help the reader identify the locations of S1 and H4. I realize the details are in Figure S5 but there are a lot of figures for the reader to digest - at the very least please refer the reader to S5 for additional detail.

We have added a label for p50 residue 210 at the start of S1 in Figures 3 and 4 to help orient the reader.

In Figure 5, is it possible to label the Zn²⁺ (1, 2 and 3?) to help the reader match each the corresponding sequence element (specified in Figure S10)). This will facilitate orientation to the overall structural features of p62 which, to my knowledge, is an unusual structure.

We have numbered the metal ions in Figure 5B, to match the sequence elements in Appendix Figure S10.

Figure S4B: Please explain in the legend what the purple color indicates.

We have clarified in Figure S4B that the dimerization domain is colored light purple, as well as explained the other shoulder and pointed end colors in S4B and S4C.

Editor accepted the manuscript.

Corresponding Author Name: Andrew P Carter

Journal Submitted to: The EMBO Journal

Manuscript Number: EMBOJ-2020-106164